# Re-ranking for image retrieval and transductive few-shot classification

Xi Shen[1], Yang Xiao[2], Shell Xu Hu[3], Othman Sbai[4], and Mathieu Aubry[5]

[1, 2, 4, 5]LIGM (UMR 8049), École des Ponts ParisTech
[3]Samsung AI Center, Cambridge

## Abstract

In the problems of image retrieval and few-shot classification, the mainstream approaches focus on learning a better feature representation. However, directly tackling the distance or similarity measure between images could also be efficient. To this end, we revisit the idea of re-ranking the top-k retrieved images in the context of image retrieval (e.g., the k-reciprocal nearest neighbors [48, 75]) and generalize this idea to transductive few-shot learning.

We propose to meta-learn the re-ranking updates such that the similarity graph converges towards the target similairty graph induced by the image labels. Specifically, the re-ranking module takes as input an initial similarity graph between the query image and the contextual images using a pre-trained feature extractor, and predicts an improved similarity graph by leveraging the structure among the involved images. We show that our re-ranking approach can be applied to unseen images and can further boost existing approaches for both image retrieval and few-shot learning problems. Our approach operates either independently or in conjunction with classical re-ranking approaches, yielding clear and consistent improvements on image retrieval (CUB, Cars, SOP, rOxford5K, rParis6K) and transductive few-shot classification (Mini-ImageNet, tiered-ImageNet and CIFAR-FS) benchmarks. Our code is available at `https://imagine.enpc.fr/~shenx/SSR/`.

## 1 Introduction

Learning deep image features that generalize beyond the training classes they have been trained on has been a clear success [63]. Using these strong features, recent works have shown that high performances can be obtained simply by computing nearest neighbors, in particular for image retrieval [1, 16, 50, 11, 30, 63] and few-shot image classification [71, 6]. In this paper we highlight that, even with these strong features, results can be further improved by a large margin through re-ranking images based on the similarity graph between neighbors. To this end, we present a graph based deep architecture for re-ranking neighborhood images via a learning approach.

Our method can be seen as revisiting and complementing classical approaches to re-ranking with deep learning techniques. These approaches can be broadly classified into query expansion that compute a new query feature based on the top retrieved neighbors [9, 8, 2, 16, 50], and k-reciprocal re-ranking that compute a new distance between images based on the Jaccard distance between confident neighbors [48, 75]. The approach we propose is related to both approaches. Since on one side, we focus successively on the neighbors of each sample, and on the other side, we update features to compute new similarities. This is achieved by designing a neural network architecture that can update the similarity graph by successively focusing on different subgraphs (as visualized in Figure 1) and updating features based on synthetic gradients [24, 21]. Our architecture for updating the similarity graph, referred to as Subgraph Similarity Refiner (SSR), operates on the adjacency matrices

35th Conference on Neural Information Processing Systems (NeurIPS 2021).

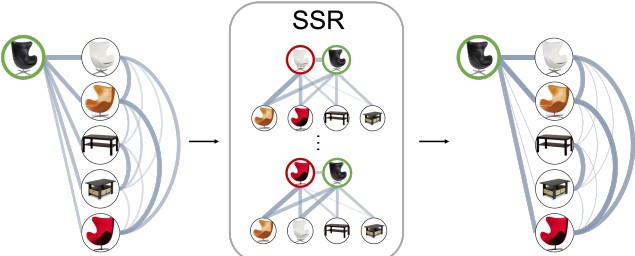

Figure 1: **Idea of our approach for image retrieval.** Given an initial similarity graph between a query and its top candidates in image retrieval (left), we propose a module dubbed Subgraph Similarity Refiner (SSR) to improve the similarity graph (right). For each sample in the dataset we build a subgraph by considering only edges with either the sample or the query (middle), order the nodes according to their similarity to the sample and predict an update for the edges of the subgraph. This idea can be applied with minor changes to transductive few-shot classification.

of the subgraphs. We focus successively on each image (as a query image), extract a subgraph, where our key technical insight is to sort all other images according to the similarity to the query image.

We show that our approach can improve different image features for both image retrieval and transductive few-shot classification. We relate both tasks by casting them as a similarity graph refinement problem, where the refined similarity graph is used for task-specific predictions. In image retrieval, based on classical re-ranking techniques query expansion [9, 8, 16, 50] and k-reciprocal [75], our method consistently improves the image retrieval performance (mean average precision, mAP) achieved by recent state-of-the-art features [11, 30, 63]. For example, applying our method with the recent ProxyNCA++ [63] features improves the mAP@R from 55.4% to 60.6% on Stanford Online Product dataset [62] and combining our approach with k-reciprocal [75] re-ranking further boost them to 62.3%. In transductive few-shot classification, we show that k-reciprocal [75] based re-ranking yields a simple but surprisingly good baseline, while by learning to re-rank our method consistently improves over several competitive transductive approaches, e.g., the synthetic information bottleneck (SIB) [21].

## 2   Approach

**Motivation**   Consider the similarity graph among $N$ images, each node of the graph corresponds to one image and each weighted edge represents the similarity between two images. The similarity graph plays an important role in computer vision. As an example, the k-nearest-neighbor classifier remains a competitive method in image classification if the similarity graph is sufficiently informative. Note that, during training, we do have access to the target similarity graph, which has similarity 1 if two images belong to the same class and 0 otherwise, *can we learn to improve an initial similarity graph of test images given that we have seen the target similarity matrix for training images?*

Indeed, in the context of image retrievel, this idea of exploring and improving the similarity graph is called *re-ranking*, which refers to the case where an initial set of images have been retrieved, we then re-rank their relevance to the query image by examining again their similarities. Certainly, this idea can be hardly scaled to a large set of images as the complexity is quadratic, but it is quite interesting in the case of few-shot learning for unseen categories / domains.

In this section, we first present an overview of our approach as a generalization to re-ranking (Section 2.1); we then introduce our model architecture, called Subgraph Similarity Refiner (SSR) (Section 2.2) and how to combine our approach with the classical k-reciprocal [75] re-ranking approach (Section 2.3); finally, in Section 2.4 we explain how we train our approach and give implementation details.

### 2.1   Learning to improve a similarity graph

We assume that we are given a set of $N$ images and for each image an initial feature. During training, we also assume that we are given a label for each image. Our goal is to predict an improved similarity, where images with the same label have a higher similarity than images with different labels. We will

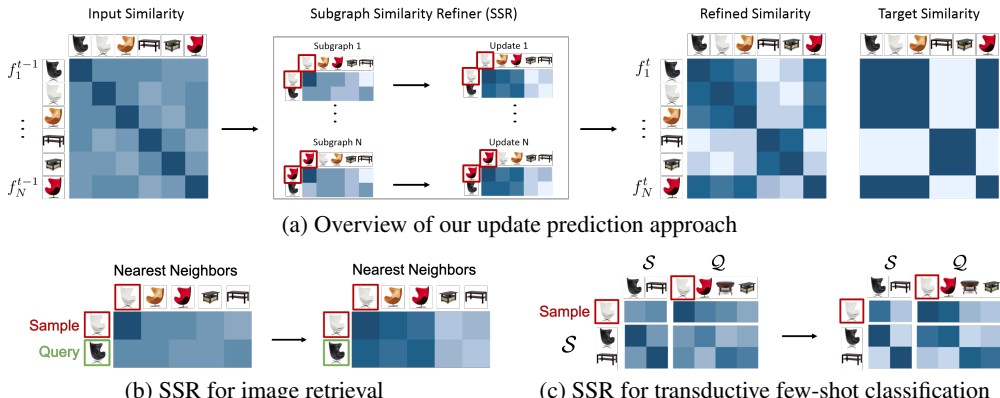

(a) Overview of our update prediction approach

(b) SSR for image retrieval  (c) SSR for transductive few-shot classification

Figure 2: **Subgraph Similarity Refiner (SSR)** learns updates for a similarity graph. It decomposes the similarity graph, that can be represented by its similarity matrix, into N subgraphs where rows and columns of the matrix are ordered depending on similarities to the subgraph reference image. The output of SSR is an improved similarity matrix. The final loss is between the predicted similarity matrix and target similarity matrix. **The details of subgraphs** are shown in (b) for image retrieval: the rows correspond to the subgraph reference image and the query image and the columns to the test images; and (c) for transductive few-shot classification: the rows correspond to the subgraph reference image and the support set $\mathcal{S}$ while the columns to the support set $\mathcal{S}$ and the query set $\mathcal{Q}$.

represent the similarity graph by its adjacency matrix, which we will refer to as similarity matrix. Note that the similarity matrix allows to explore neighbors of the nearest neighbors (high order nearest neighbors), which is a classical and effective way to tackle the re-ranking problem [48, 75].

More formally, for $i \in \{1, \ldots, N\}$ we denote by $f_i^0$ the initial feature of $i^{th}$ image and $y_i$ its label. The initial similarity matrix is given by:

$$\mathbf{S}^0 = \left[s_{ij}^0\right]_{i,j \in \{1,\ldots,N\}} \quad \text{with} \quad s_{ij}^0 = \frac{\langle f_i^0, f_j^0 \rangle}{\|f_i^0\| \|f_j^0\|}. \tag{1}$$

Our goal is to predict the target similarity matrix $\hat{\mathbf{S}} = \left[\mathbb{1}_{y_i=y_j}\right]_{i,j \in \{1,\ldots,N\}}$ where $\mathbb{1}_{y_i=y_j}$ is the indicator function of $y_i = y_j$. We refine the similarity matrix iteratively. First, consider the following update given a similarity matrix $\mathbf{S}^{t-1}$:

$$\tilde{\mathbf{S}}^t := \left[\tilde{s}_{ij}^t\right]_{i,j \in \{0,1,\ldots,N\}} = \mathbf{S}^{t-1} + \mathrm{G}(\mathbf{S}^{t-1}), \tag{2}$$

where G is the Subgraph Similarity Refiner (SSR) that we will describe in the next section. The issue with such an update is the flexibility. We therefore would like to constrain the updated similarity matrix to have the form

$$\mathbf{S}^t = \left[s_{ij}^t\right]_{i,j \in \{1,\ldots,N\}} \quad \text{with} \quad s_{ij}^t = \frac{\langle f_i^t, f_j^t \rangle}{\|f_i^t\| \|f_j^t\|}. \tag{3}$$

This amounts to construct the update in features rather than in the similarity matrix directly. As such, $\mathbf{S}^t$ always remain positive semidefinite. Now, the question is how we make use of $\tilde{\mathbf{S}}^t$ to obtain $f^t$. Given $f^{t-1}$, in fact, we only need to move a single gradient step along the projection direction according to the Euclidean distance $L_f(f) := \frac{1}{2}\sum_{i,j}(\tilde{s}_{i,j}^t - \frac{\langle f_i, f_j \rangle}{\|f_i\| \|f_j\|})^2$. Specifically, we obtain $f^t$ by considering $\tilde{\mathbf{S}}^t$ as an intermediate result:

$$\forall i \colon f_i^t = f_i^{t-1} - \lambda \frac{\partial L_f}{\partial f_i}(f^{t-1}) \quad \text{with} \quad \frac{\partial L_f}{\partial f_i}(f^{t-1}) = -\sum_j (\tilde{s}_{i,j}^t - s_{i,j}^{t-1}) \frac{\partial s_{i,j}^{t-1}}{\partial f_i^{t-1}}(f^{t-1}) \tag{4}$$

where $\lambda$ is the step size. Indeed, this update can be seen as applying the synthetic gradient descent [21, 24] on the features. Here, we motivate it from a different perspective.

Note that the network G defines a meta-model, as the update is supposed to generalize to unseen image categories. We learn G by minimizing a task-specific loss between the target similarity matrix $\hat{\mathbf{S}}$ and the updated similarity matrix $\mathbf{S}^T$. The key for this approach to work is the architecture design of the network G, which we detail in the next section.

## 2.2 Subgraph Similarity Refiner (SSR)

The adjacency matrix $\mathbf{S}$ is a natural way to encode a similarity graph. However, it is defined up to the order of the nodes, i.e. permutations of its column and rows. For this reason, learning directly to make a prediction from the similarity matrix is challenging. One can simply order the nodes with respect to their distance to a specific sample, such as the query in the case of image retrieval, but this gives one of the nodes a specific role and it might be hard for the network to use the similarity structure among the other nodes, especially since $\mathbf{S}$ is very high dimensional and overfitting might be an issue. Thus, we propose a permutation-invariant architecture illustrated in Figure 2a.

Our key idea is to make $N$ update predictions by considering only subgraphs centered on each node $i$ and then to aggregate them to obtain an update prediction on the full graph. We can associate a similarity matrix $\mathbf{M}_i$ to the $i^{th}$ subgraph, where we order nodes according to their similarity to the node $i$. Note that the order will be different for each subgraph. We then predict updates $\mathrm{g}(\mathbf{M}_i)$ for each of the subgraphs using a simple network $\mathrm{g}$ . Finally, we aggregate the predictions on all the subgraphs by summing the updates predicted for each subgraph at the relevant position in $\mathbf{S}$:

$$\mathrm{G}(\mathbf{S}) = \mathrm{GraphSum}\left(\{\mathrm{g}(\mathbf{M}_i)\}_{i=1}^N\right) \tag{5}$$

where $\mathrm{g}$ is a multi-layer perceptron (MLP) and GraphSum is an aggregation operator to account for the summation with respect to graph structure since each node is contained in multiple subgraphs.

We now discuss the exact structure of the subgraph and associated similarity matrix $\mathbf{M}_i$ that we use in the case of image retrieval or transductive few-shot classification. The choice of structures in both cases are validated by the ablation studies presented in Section 3.3.

**Image Retrieval.** In image retrieval, we search for images similar to a *query image* in a pool of *test images*. We assume that a first algorithm already selected the $N-1$ test images most similar to the query and we focus on improving the similarities between the resulting set of $N$ images (the query and its $N-1$ nearest neighbors). By increasing the similarities between the query and the positive images against those of the negatives, we aim to obtain a better retrieval result where positives should be ranked before negatives for the $N-1$ test images selected for the query. During training, we optimize the InfoNCE loss [44] with a learnable temperature parameter $\tau$:

$$\sum_{i,j:y_i=y_j=y_{qry}} -\log\left(\frac{\exp(\tau s_{ij}^T)}{\exp(\tau s_{ij}^T) + \sum_{k:y_k \neq y_i}\exp(\tau s_{ik}^T)}\right) \tag{6}$$

where $s_{ij}^T$ signifies the similarity between the $i^{th}$ sample and $j^{th}$ sample after $T$ similarity updates.

As visualized in Figure 2b, we build the $i^{th}$ subgraph by considering only edges that connect nodes to either the $i^{th}$ sample or the query. Rather than considering $\mathbf{M}_i$ as the adjacency matrix of this graph, we define it as a $2 \times N$ matrix defined as follows. Each column corresponds to a different sample and the samples are ordered with respect to their similarity of the $i^{th}$ sample. The values in the first row are the similarities between the $i^{th}$ sample and the samples associated to each column, it is thus decreasing. The values in the second row are the similarity between the query and the samples associated with each column. Note that for the subgraph corresponding to the query, the two lines of the matrix are the same and actually correspond to the same edges in the graph.

**Transductive few-shot classification.** In few-shot classification, the set of $N$ images to consider can be divided in two: a support set $\mathcal{S}$ with known labels and an *unlabelled* query set $\mathcal{Q}$ for which we want to predict labels. Our approach assumes that $\mathcal{Q}$ is accessible during the inference. This setting is known as transductive few-shot classification. For each sample in the query set, we aim to increase the similarities between itself and the support images from the same class comparing to those from different classes. To this end, we minimize the Cross Entropy loss:

$$\sum_{i \in \mathcal{Q}, j \in \mathcal{S}:y_i^q=y_j^s} -\log\left(\frac{\exp(\tau s_{ij}^T)}{\exp(\tau s_{ij}^T) + \sum_{k \in \mathcal{S}, y_i^q \neq y_k^s}\exp(\tau s_{ik}^T)}\right) \tag{7}$$

where $\tau$ is a learnable temperature parameter, $y^q$ and $y^s$ are labels of query and support samples.

As visualized in Figure 2c, we build the $i^{th}$ subgraph by considering only the edges that connect nodes either to the $i^{th}$ sample or to the support set. Similar to the case of retrieval, we represent this

graph using a structured matrix, but we keep the nodes corresponding to the support and query set separated. We define $\mathbf{M}_i$ as a $(|\mathcal{S}| + 1) \times (|\mathcal{S}| + |\mathcal{Q}|)$ matrix with $|\cdot|$ the number of elements in a set. Its first $|\mathcal{S}|$ columns and last $|\mathcal{Q}|$ columns correspond to the samples in support set and query set, respectively. They are both sorted by decreasing similarity with respect to the $i^{th}$ sample. While the values in the first row of $\mathbf{M}_i$ represent the similarities between the $i^{th}$ sample and the samples in different columns, the last $|\mathcal{S}|$ rows represent similarities between support samples and different columns. Note that if the $i^{th}$ sample comes from the support set, one row will be repeated in $\mathbf{M}_i$.

## 2.3 Combination with k-reciprocal distance [75]

Instead of taking CNN feature similarities as input of G, our approach can be augmented with other distances. In particular, we show that it leads to stronger results when using the k-reciprocal feature distance [75]. We summarize in this section how the k-reciprocal feature distance is obtained and how we update it with our approach.

The k-reciprocal feature is computed from k-reciprocal neighbors [19]. Writing $\mathtt{top(a,k)}$ the k nearest neighbors of the feature $\mathtt{a}$, the set of k-reciprocal neighbors $\mathcal{R}(\mathtt{a}, \mathtt{k})$ of $\mathtt{a}$ is defined as :

$$\mathcal{R}(\mathtt{a}, \mathtt{k}) = \{\mathtt{b} | \mathtt{b} \in \mathtt{top(a,k)} \cap a \in \mathtt{top(b,k)}\}. \tag{8}$$

The backward verification aims at reducing the number of false matches in the k-reciprocal neighbors. To consider potential positive matches excluded from the k nearest neighbors, [75] proposed to add the $\frac{k}{2}$-reciprocal neighbors of $\mathtt{b} \in \mathcal{R}(\mathtt{a}, \mathtt{k})$ into an expanded set $\mathcal{R}^*(\mathtt{a}, \mathtt{k})$ if $\mathcal{R}(\mathtt{b}, \frac{k}{2})$ shares enough neighbors with $\mathcal{R}(\mathtt{a}, \mathtt{k})$.

The final proposed distance $d$ in k-reciprocal [75] is a combination of the euclidean distance between normalized features and the Jaccard distance $d_J$ computed with the expanded sets :

$$d(\mathtt{a}, \mathtt{b}) = \alpha \left\| \frac{\mathtt{a}}{\|\mathtt{a}\|} - \frac{\mathtt{b}}{\|\mathtt{b}\|} \right\|^2 + (1 - \alpha) d_J(\mathcal{R}^*(\mathtt{a}, \mathtt{k}), \mathcal{R}^*(\mathtt{b}, \mathtt{k})) \tag{9}$$

where $\alpha$ is a hyper-parameter representing the contribution of the feature similarity. For more details about the Jaccard distance, we refer to [75].

Now, we compute the distance matrix $\mathbf{J}$ corresponding to the Jaccard distance obtained from the initial features and consider it as fixed. We then consider the distance defined by Equation 9 to build our graphs updating only the feature similarity in the first part of the equation. More precisely, at each iteration $t$, we use $\mathbf{D}^t = 2\alpha(1 - \mathbf{S}^t) + (1 - \alpha)\mathbf{J}$ as input of the SSR, and consider its output is an update to $\mathbf{S}^t$ only and the final objective function remains on $\mathbf{S}^T$. We also tried to use the final objective on $\mathbf{D}^T$, but observed a severe overfitting on the training set and worse results.

## 2.4 Network architecture and implementation details

**Architecture.** Each subgraph update in our SSR module is performed by a three-layer perceptron with constant hidden-layer size 1,024 for image retrieval and 4,096 for few-shot classification. Further increasing the model size leads to similar performances. All the layers except the last one are followed by ReLU activations and Instance Normalization [64], which we also apply to the input matrix.

**Optimization.** We optimize our networks using SGD with momentum 0.9. The batch size is set to 1 since there are numerous images to consider in a single similarity graph and increasing the batch size does not improve the performance. For image retrieval, we use a single update of the model ($T = 1$) and training converges in 10K iterations with a fixed learning rate of 1e-5. Larger $T$ leads to similar performance. The analysis of $T$ and $\lambda$ on rOxford5K [49] and rParis6K [49] are available in the supplimentary material. The entire training on CUB [67] takes 6 hours on a single GeForce 1080 Ti GPU. For few-shot classification, we first train for 30K iterations with $T = 1$: the learning rate is set to 0.1 for 5K iterations then to 0.01 for another 25K iterations. Then, keeping a learning rate of 0.01, we train for 10K iterations with $T = 2$ and 10K more with $T = 3$. We find that $T = 3$ leads to the most stable improvement and include this analysis in the supplementary material. The whole training process on mini-ImageNet [66] takes 20 hours on a single GeForce 1080 Ti GPU.

| Method \Feature | CUB [67], mAP@R | | | CARS [31], mAP@R | | | SOP [62], mAP@R | | | rOxford5K [49], mAP | | rParis6K [49], mAP | |
|---|---|---|---|---|---|---|---|---|---|---|---|---|---|
| | GL [11] | PA [30] | PNCA++ [63] | GL [11] | PA [30] | PNCA++ [63] | GL [11] | PA [30] | PNCA++ [63] | Medium [17] | Hard [17] | Medium [17] | Hard [17] |
| LAttQE [17] | - | - | - | - | - | - | - | - | - | 73.4 | 49.6 | 86.3 | 70.6 |
| Region Diffusion [23, 49]† | - | - | - | - | - | - | - | - | - | 69.0 | 44.7 | 89.5 | 80.0 |
| Feat. only | 24.5 | 27.0 | 29.6 | 27.8 | 28.3 | 33.2 | 46.9 | 51.0 | 55.4 | 67.3 | 44.3 | 80.6 | 61.5 |
| Feat + SSR | 34.0 | 35.5 | 39.5 | 38.3 | 38.7 | 45.8 | 50.9 | 54.8 | 60.6 | 75.6 | 54.2 | 84.4 | 67.8 |
| AQE [9] | 28.3 | 31.2 | 34.1 | 34.4 | 34.9 | 40.9 | 49.3 | 54.8 | 58.4 | 70.8 | 48.0 | 85.3 | 68.8 |
| AQE + Ours | 34.1 | 36.1 | 40.0 | 40.3 | 39.5 | 48.3 | 49.5 | 53.8 | 60.5 | 74.2 | 54.4 | 84.5 | 68.7 |
| AQEwD [16] | 28.4 | 31.4 | 34.2 | 34.6 | 35.1 | 40.9 | 49.3 | 55.0 | 58.6 | 70.8 | 48.0 | 84.5 | 67.6 |
| AQEwD + SSR | 34.0 | 35.9 | 41.2 | 39.2 | 39.7 | 48.4 | 49.9 | 54.2 | 60.5 | 74.1 | 54.1 | 83.9 | 67.9 |
| DQE [2] | 26.4 | 30.6 | 33.0 | 31.8 | 34.5 | 37.0 | 48.7 | 55.0 | 58.3 | 69.5 | 46.1 | 84.0 | 67.0 |
| DQE + SSR | 33.1 | 35.5 | 39.9 | 39.4 | 39.0 | 46.2 | 49.8 | 53.8 | 60.4 | 73.6 | 53.7 | 83.1 | 67.0 |
| $\alpha$QE [50] | 28.3 | 31.2 | 34.1 | 34.5 | 34.5 | 40.9 | 49.3 | 54.8 | 58.4 | 68.2 | 44.0 | 84.3 | 67.2 |
| $\alpha$QE + SSR | 33.6 | 34.7 | 40.7 | 40.4 | 35.7 | 48.4 | 49.7 | 54.0 | 60.4 | 71.6 | 51.1 | 83.5 | 67.3 |
| k-reciprocal [75]† | 37.6 | 41.8 | 48.1 | 49.9 | 50.2 | 58.5 | 51.7 | 56.3 | 61.7 | 72.1 | 50.7 | 87.9 | 74.8 |
| k-reciprocal [75]† + SSR | 38.3 | 42.3 | 49.8 | 51.1 | 50.9 | 60.4 | 52.5 | 56.6 | 62.3 | 75.0 | 53.1 | 88.7 | 75.9 |

† carries the extra cost of the graph over the whole dataset.

Table 1: **Image retrieval.** For [67], CARS [31] and SOP [62], we use three features: GL [11], PA [30] and PNCA++ [63], and report mAP@R, which follows [40]. For rOxford5K [49] and rParis6K [49], we use the feature provided in [17] and report mAP, which follows [17].. The best and the 2nd best results are in red and blue respectively.

## 3 Experiments

In this section, we cover our experimental setups and results for image retrieval and few-shot image classification. Since these two problems are different in data processing and performance evaluation, we separate the discussions into two sub-sections followed by a joint ablation study.

### 3.1 Image retrieval

**Datasets.** We consider five image retrieval datasets, namely, CUB [67], CARS [31], SOP [62], rOxford5K [49] and rParis6K [49]. For CUB, CARS, SOP, we follow the standard split [11]: for CUB, the first 100 species (5,864 images) are used for training and the remaining 100 species (5,924 images) are used for testing; for CARS, the first 98 classes (8,054 images) are used for training and the other 98 classes (8,131 images) are kept for testing; for SOP, the dataset is separated into 11,318 training classes (59,551 images) and 11,316 testing classes (60 502 images). For rOxford5K [49] and rParis6K [49], we follow [17] and use the dataset SFM120k [50] , which is built with structure-from-motion pipeline, and clusters for the same 3D scene are cast as categories. We take features in [17], which already leads to good performance on the training set. For most training samples, the mAPs on the training set are already quite high and training SSR using the raw nearest neighbors makes it perform well only for high mAP queries. To address this problem, we sample only difficult examples. To combine other re-ranking methods and SSR, we directly apply the trained SSR to the top retrieved samples given by other re-ranking methods. More details can be found in the supplementary material.

**Evaluation metric.** For CUB, CARS, SOP, following [40], we choose the Mean Average Precision at $R$ (MAP@$R$) as our main evaluation metric: MAP@$R = \frac{1}{R} \sum_{i=1}^{R} P(i)$ with $R$ the total number of true positive samples and $P(i)$ is the precision at $i$ if the $i^{th}$ retrieval is correct and 0 otherwise. For rOxford5K and rParis6K, we follow [17] and report stand Mean Average Precision (mAP) on medium and hard queries.

**Baselines.** We conduct exhaustive experiments with three recent feature representations: a) features trained with group loss [11] (GL), b) features trained with proxy anchor loss [30] (PA) and c) features trained with proxy neighborhood component analysis method [63] (PNCA++). For each feature, we report the baseline results obtained from the k-reciprocal method [75] and four standard query expansion methods [17]: Average Query Expansion (AQE, [9, 16, 50]), Average Query Expansion with Decay (AQEwD, [50]), Alpha Query Expansion ($\alpha$QE [50]) and Discriminative Query Expansion (DQE [2]). The baselines and details are provided in the supplementary material.

**Results.** Our results are shown in Table 1. Applying our SSR on original features ("*Feat. + SSR*") largely improves the retrieval performance of using original features ("*Feat.*") on different datasets for different metric. In almost all cases our method alone outperforms all the query expansion baselines and results can be further improved by applying it to the query-expanded results. The performance improvement brought by k-reciprocal [75] should be attributed to the statistics of neighbors of neighbors, while our method can further boost upon that. These results suggest that our method can be combined with many feature extractors for image retrieval to attain better performance. Note that the choices of N are available in the supplementary material.

| Methods | Trans. | Backbone | mini-ImageNet [66] | | tiered-ImageNet [51] | | CIFAR-FS [45] | |
|---|---|---|---|---|---|---|---|---|
| | | | 1-shot | 5-shot | 1-shot | 5-shot | 1-shot | 5-shot |
| MatchNet [66] | | Conv-4-64 | 44.2 | 57.0 | – | – | – | – |
| ProtoNet† [60] | | Conv-4-64 | 49.4±0.8 | 68.2±0.7 | 53.3±0.9 | 72.7±0.7 | 55.5±0.7 | 72.0±0.6 |
| GNN [13] | | Conv-4-64 | 50.3 | 66.4 | 61.9 | 75.3 | – | – |
| Gidaris et al. [14] | | Conv-4-64 | 54.8±0.4 | 71.9±0.3 | – | – | 63.5±0.3 | **79.8±0.2** |
| MAML‡ [12] | BN | Conv-4-64 | 48.7±1.8 | 63.1±0.9 | 51.7±1.8 | 70.3±1.8 | – | – |
| TPN [35] | ✓ | Conv-4-64 | 55.5±0.9 | 69.9±0.7 | 59.9±0.9 | 73.3±0.8 | – | – |
| SIB [21] | ✓ | Conv-4-64 | 58.0±0.6 | 70.7±0.4 | – | – | 68.7±0.6 | 77.1±0.4 |
| EPNet [53] | ✓ | Conv-4-64 | 59.3±0.9 | 73.0±0.6 | 60.0±1.0 | 73.9±0.8 | – | – |
| Baseline | | Conv-4-64 | 52.4±0.4 | 69.6±0.4 | 55.2±0.5 | 72.3±0.4 | 57.8±0.5 | 75.3±0.4 |
| + k-reciprocal [75] | ✓ | Conv-4-64 | 58.6±0.7 | 72.2±0.5 | 63.1±0.8 | **75.0±0.6** | 66.6±0.8 | 78.1±0.6 |
| + Ours | ✓ | Conv-4-64 | **62.1±0.6** | **73.2±0.4** | **65.1±0.6** | 74.1±0.5 | **72.0±0.6** | 78.5±0.4 |
| TADAM [45] | | ResNet-12 | 58.5±0.3 | 76.7±0.3 | – | – | – | – |
| MetaOptNet-RR [33] | | ResNet-12 | 61.4±0.6 | 77.9±0.5 | 65.4±0.7 | 81.3±0.5 | 72.6±0.7 | **84.3±0.5** |
| ProtoNet+MABAS [28] | | ResNet-12 | 65.1±0.9 | **82.7±0.5** | – | – | 73.5±0.9 | **85.5±0.7** |
| EGNN* [29] | ✓ | ResNet-12 | 64.0 | 77.2 | 66.5 | 82.5 | – | – |
| CAN [20] | ✓ | ResNet-12 | 67.2±0.6 | 80.6±0.4 | 73.2±0.6 | 84.9±0.4 | – | – |
| EPNet [53] | ✓ | ResNet-12 | 66.5±0.9 | **81.1±0.6** | 76.5±0.9 | **87.3±0.6** | – | – |
| Baseline | | ResNet-12 | 57.6±0.5 | 73.5±0.4 | 68.8±0.5 | 83.5±0.4 | 66.4±0.5 | 80.4±0.4 |
| + k-reciprocal [75] | ✓ | ResNet-12 | **67.3±0.7** | 78.0 ±0.5 | **77.3±0.8** | 85.7±0.5 | **73.6±0.8** | 82.1±0.5 |
| + Ours | ✓ | ResNet-12 | **68.1±0.6** | 76.9±0.4 | **81.2±0.6** | 85.7±0.4 | **76.8±0.6** | 83.7±0.4 |
| LEO [55] | | WRN-28-10 | 61.8±0.1 | 77.6±0.1 | 66.3±0.1 | 81.4±0.1 | – | – |
| Gidaris et al. [14] | | WRN-28-10 | 62.9±0.5 | 79.9±0.3 | 70.5±0.5 | **85.0±0.4** | 76.1±0.3 | **87.8±0.2** |
| SIB‡‡ [21] | ✓ | WRN-28-10 | 70.0±0.6 | 78.9±0.4 | 72.9* | 82.8* | **80.0±0.6** | 85.3±0.4 |
| SIB+E³BM [36] | ✓ | WRN-28-10 | **71.4** | **81.2** | 75.6 | 84.3 | – | – |
| EPNet [53] | ✓ | WRN-28-10 | 70.7±0.9 | **84.3±0.5** | **78.5±0.9** | **88.4±0.6** | – | – |
| Baseline | | WRN-28-10 | 61.9±0.5 | 77.8±0.3 | 69.4±0.5 | 83.4±0.4 | 69.5±0.5 | 83.5±0.4 |
| + k-reciprocal [75] | ✓ | WRN-28-10 | 68.1±0.8 | 79.4±0.5 | 76.4±0.7 | 84.8±0.5 | 76.7±0.8 | 84.9±0.5 |
| + Ours | ✓ | WRN-28-10 | **72.4±0.6** | 80.2±0.4 | **79.5±0.6** | 84.8±0.4 | **81.6±0.6** | **86.0±0.4** |

†Results from [33]. ‡Results from [35]. *Results from [36].
‡‡ we use the same pre-trained features of WRN and Conv-4-64 as [21] on mini-ImageNet.

Table 2: **Transductive few-shot classification.** 5-way few-shot classification accuracies (%) on mini-ImageNet [66], tiered-ImageNet [51], and CIFAR-FS [4]. We report average classification accuracy (with 95% confidence intervals) over 2000 episodes on the test set. We highlight in grey the results of our baseline features with nearest neighbor classifier (entry "*Baseline*"), with k-reciprocal [75] re-ranking (entry "+ *k-reciprocal*"), and with our approach (entry "+ *Ours*"). The best and the 2nd best results are in red and blue respectively.

## 3.2 Transductive few-shot classification

**Dataset.** We evaluate our approach on three standard few-shot classification datasets: mini-ImageNet [66], tiered-ImageNet [51], and CIFAR-FS [4]. mini-ImageNet and tiered-ImageNet contain a subset of ImageNet images resized to 84×84. mini-ImageNet contains 100 classes and 600 images per class. It is split into 64 classes for training, 16 for validation and 10 for testing. tiered-ImageNet contains a larger subset of ImageNet with 608 classes and 1 300 images per class. It is split into 351 classes for training, 97 for validation and 160 for testing. CIFAR-FS was created by dividing the original CIFAR-100 [32] into 64 training classes, 16 validation classes and 20 testing classes. Each class has 600 images. The image resolution is 32×32.

**Architectures and baseline features.** We experimented with three architectures: WRN-28-10 [74, 15, 14, 21], ResNet-12 [38, 45, 33] and Conv-4-64 [15, 14, 21]. WRN-28-10 is commonly evaluated in few-shot classification [74, 15, 14, 21]. ResNet-12 is the architecture used by [38, 45, 33]. Conv-4-64 is widely used in few-shot learning [15, 14, 21] and has 4 convolutional modules, with 3 × 3 convolutions, followed by Batch Normalization [22], ReLU non-linearity and 2 × 2 Max Pooling. For all architectures and datasets, we use a baseline feature obtained by pre-training a cosine classifier [15] to initialize our approach. Note that this pre-training is carried out on the train-set with hyper-parameter selection on the validation set. The cosine classifier is trained following a standard training strategy: we use the SGD optimizer with momentum 0.9 and batch size 64 for 120 epochs. The first 50 epochs are with learning rate 0.1, the next 50 with learning rate 0.01, and the last 20 with learning rate 0.001. We adopt standard data augmentation: resizing, cropping and horizontal flipping.

**Results.** We compare our approach to state-of-the-art methods in Table 2. For all the datasets and backbones, we report the performance of the baseline features ('*Baseline*'), k-reciprocal [75] with the baseline feature ('+ *k-reciprocal*') and our approach with the baseline features ('+ *Ours*').

Interestingly, a simple baseline by combining the baseline feature and k-reciprocal, without any learning procedure, achieves comparable performance with recent methods [21, 36, 53] on all three datasets. Moreover, our method achieves the best performance on 1-shot classification and obtains competitive performance on 5-shot classification. In particular, compared to SIB [21], which is also an approach using synthetic gradient and similar baseline features, our approach yields a clear and consistent improvement. Note that combining k-reciprocal and our method did not bring any improvement over our method alone for few-shot classification (see results in supplementary material).

### 3.3 Ablation study

**Image retrieval.** We provide an analysis on CARS [31] with using Group Loss features [11] and N = 100 to analyze our approach for image retrieval. The results are presented in Table 3. This ablation provides several insights. First, using subgraphs is essential: performance simply using as input and predicting the full similarity matrix (*w.o subgraph*) leads to performances close to the baseline. Second, sorting the columns in each subgraph is important, because it makes the information about each sample easier for the network to extract, as can be seen in the 1.8% performance loss when the columns are sorted according to their similarity with the query (*w.o local sort*). Third, the feature update (*w.o feat. update*), which allows to leverage the input features, is also important, removing it degrades the performance by 3.8%. Finally, adding or removing rows in $\mathbf{M}_i$ shows that using the subgraphs we define is a good trade-off: removing connections to the $i^{th}$ sample (*w.o sample*) or the query (*w.o query*) degrades performance, and so does considering the full graph (*w.o other*). Note that independently of the graph they represent, the matrices $\mathbf{M}_i$ are still focused on sample $i$ which is used to sort their rows and columns.

| Method | mAP@R |
|---|---|
| Baseline (features) | 27.8 |
| Ours (sample and query) | 34.0 |
| w.o sample | 32.9 |
| w.o query | 28.2 |
| w. other | 33.0 |
| w.o local sort | 32.2 |
| w.o subgraph | 27.9 |
| w.o feat. update | 30.2 |

(a) mAP@R on CARS [31]

(b) Variants of $\mathbf{M}_i$

Table 3: **Image retrieval:** Ablation study on CARS [31].

**Transductive few-shot classification.** We now present a similar analysis on transductive few-shot classification. The results are in Table 4. We identify variants of the subgraph similarity matrices $\mathbf{M}_i$ by writing the samples we use in rows and columns. First, note that the optimal choice is the one presented in Section 2.2, using matrices $\mathbf{M}_i$ with rows corresponding the $i^{th}$ sample and the support set $\mathcal{S}$ and columns corresponding to the support set $\mathcal{S}$ and the query set $\mathcal{Q}$. Second, the local sorting is more important for few-shot classification than for image retrieval, removing it (*w.o local sort*) degrades the performance by 9%. The reason is that in image retrieval we could order images with respect to their similarities to the query, while in few-shot classification there is no natural order. Third, similar to image retrieval, the decomposition into subgraphs (*w.o subgraph*) is crucial to obtain improved performance. Finally, updating features (*w.o feat. update*) is also important in this task and brings an extra 1.4% improvement.

| Method | $\mathbf{M}_i$ Rows | $\mathbf{M}_i$ Col. | Acc % |
|---|---|---|---|
| Baseline (features) | | | 61.9 |
| Ours | sample, $\mathcal{S}$ | $\mathcal{S}, \mathcal{Q}$ | 72.4 |
| | sample, $\mathcal{S}, \mathcal{Q}$ | $\mathcal{S}, \mathcal{Q}$ | 72.2 |
| | $\mathcal{S}$ | $\mathcal{S}, \mathcal{Q}$ | 72.0 |
| | sample | $\mathcal{S}, \mathcal{Q}$ | 70.8 |
| | sample, $\mathcal{S}$ | $\mathcal{Q}$ | 71.7 |
| | sample, $\mathcal{S}$ | $\mathcal{S}$ | 66.3 |
| w.o local sort | sample, $\mathcal{S}$ | $\mathcal{S}, \mathcal{Q}$ | 63.4 |
| w.o subgraph | sample, $\mathcal{S}$ | $\mathcal{S}, \mathcal{Q}$ | 63.2 |
| w.o feat. update | sample, $\mathcal{S}$ | $\mathcal{S}, \mathcal{Q}$ | 71.0 |

(a) 1-shot accuracy

(b) Variants of $\mathbf{M}_i$

Table 4: **Transductive few-shot classification:** Ablation study on mini-ImageNet [66] with WRN [74].

## 4 Related Work

**Image Retrieval.** Given a query image, image retrieval searches for similar images in a set of test images. Traditional approaches [59, 37, 42, 46, 25, 2] seek to match a bag-of-words representation. Recent approaches rather focus on learning a good global representation of images [16, 50, 52] to map similar samples closer to each other against dissimilar ones, which is also addressed by deep metric learning. Pair based loss has a long story in metric learning. Contrastive loss [5, 7, 18, 50] pulls positive pairs together and pushes negatives far apart. Triplet loss [72, 68, 57, 16] enforces the

distance between of a positive pair to be smaller than a negative pair. N-pair [61] and Lifted Structure loss [43] associate an anchor with a positive but multiple negatives. Both Ranked List [70] and Multi-Similarity loss [69] take into account all positive and negative pairs in a batch and Multi-Similarity assigns a different weight to each pair. Instead of optimizing on pairs of samples, proxy based losses consider distances between samples and learnable proxies [39, 73, 47, 3, 63]. Proxy-NCA [39] allows to reduce computation by approximating NCA [54] loss with proxies. Proxy Anchor [30] improves Proxy-NCA [39] by assigning a proxy for each class and associating each proxy with the entire batch. ProxyNCA++ [63] further improves ProxyNCA and proposes several enhancements, such as low temperature scaling, Global Max Pooling, etc.

**Few-shot classification.** Few-shot learning [66, 60, 12] refers to learning from a few annotated examples. Recent works are more focused on meta-learning [56] which aims at learning the ability to solve new tasks from previous experiences. Our work focuses in particular on the transductive setting, which differs from the standard few-shot learning by assuming the unlabelled query set to be accessible during inference. By sharing information between test examples through Batch Normalization [22], MAML [12, 41] is the first method to apply transduction. The goal of MAML is to learn a parameter initialization that can be fine-tuned quickly on a new task. Reptile [41] approximates MAML with first-order derivatives rather than second-order derivatives. In the same spirit of MAML, SIB [21] proposed to adapt to a new task by learning to predict synthetic gradient [24]. SIB can be further improved with the ensemble of epoch-wise empirical Bayes models [36]. CAN [20] designs a cross attention module to adaptively extract support and query features. TPN [35] learns to propagate labels with a neural network. EGNN [29] takes into account both node and edge feature to update the graph. EPNet [53] proposed to use embedding propagation as an unsupervised regularizer for manifold smoothing. While being related to graph based approaches, our method is different to the previous works in the sense that the graph update is done in an implicit way learned automatically by the network, rather than an explicitly designed scheme.

**Re-ranking for image retrieval.** Image retrieval results are usually improved by re-ranking the nearest neighbors [9, 16, 50, 75]. Query expansion is a classic re-ranking approach that constructs an expanded query from the top-k retrieved samples. The expanded query can be a linear combination of the top-k retrievals (AQE, AQEwD, alphaQE) [9, 8, 16, 50], or as proposed in [2], a linear SVM classifier for each query trained using top ranked features as positives, and low ranking features as negatives. Recently, LAttQE [17] proposed to apply several successive explicit aggregators with self-attention [65] to obtain the updated features. Another line of research focuses on exploring higher-order neighbors [34]. The methods can either use explicitly label propagation on the nearest neighbor graph [77, 76, 10, 23], or directly encode neighbor information into image descriptors [58, 34]. Shen et al. [58] proposed to encode images by their k nearest neighbors. The final rank of a database image is determined by its ranks in the retrieval results of the query and the query's k nearest neighbors. The k-reciprocal nearest neighbors is used in [26, 27] by taking into account the symmetry of k-neighborhood relationship to update distances. The concept of k-reciprocal nearest neighbors is formally presented in [48] where two images are k-reciprocal nearest neighbors if both are nearest neighbors of each other. This idea is then further explored and demonstrated to be effective for person re-identification in [75].

## 5   Conclusion

In this work, we presented a deep approach for image re-ranking and an architecture specifically designed to update similarity graphs. We apply it to two problems that are usually tackled by different methods: image retrieval and transductive few-shot classification. Experimental results suggested that our approach can be applied alone or be complementary to classical re-ranking methods in image retrieval and both lead to significant improvements. In transductive few-shot classification, we showed that applying a classical re-ranking method to pre-trained features leads to strong results and we can attain state of the art with our approach.

**Acknowledgment** This work was supported in part by ANR project EnHerit ANR-17-CE23-0008, project Rapid Tabasco, and IDRIS under the allocation AD011011160R1 made by GENCI.

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
