# Supplementary material
# Re-ranking for image retrieval and transductive few-shot classification

Xi Shen[1], Yang Xiao[2], Shell Xu Hu[3], Othman Sbai[4], and Mathieu Aubry[5]

[1, 2, 4, 5]LIGM (UMR 8049), École des Ponts ParisTech
[3]Samsung AI Center, Cambridge

## 1   Organization

This document contains the following sections:

- Details on rOxford5K and rParis6K are in Section 2, which includes: i) the detail of hard training data sampling, ii) dependency on $T$ and $\lambda$ (Section 2.1), iii) analysis of hyper-parameters in QE methods (Section 2.2), iv) analysis of hyper-parameters in k-reciprocal (Section 2.3).

- The detailed results on CUB, CARS and SOP including other metrics such as Recall and PR, and results with different number of neighbors N are provided in Section 3.

- The results of query expansion baselines on CUB, CARS and SOP with grid search on hyper-parameters are in Section 4.

- The results of k-reciprocal [20] on CUB and CARS with grid search on hyper-parameters are in Section 5.

- The dependency on the number of updates T for 1-shot classification on Mini-ImageNet [17] is provided in Section 6

- The 1-shot results of combining our approach and k-reciprocal [20] is provided in Section 7

## 2   Details on rOxford5K and rParis6K

**Hard training data sampling**   The main difficulty is that there is no standard clean training set. One choice is SFM120k used in [14], which is built with structure-from-motion pipeline, and clusters for the same 3D scene are cast as categories. We take features in [7], which already leads to good performance on the training set. For most training samples, the mAPs on the training set are already quite high and training SSR using the raw nearest neighbors makes it perform well only for high mAP queries. To address this problem, we sample only difficult examples: for each query, we sample 1K database images, the query and its nearest neighbors will be training samples only if the mAP is not saturated ($\leq 0.8$) and there are sufficient true positive samples present in the nearest neighbors ($\geq 5$).

### 2.1   Dependency on the number of updates $T$ and step size $\lambda$

In Table 1, we provide the analysis of hyper-parameters in our SSR on rOxford5K and rParis6K. Similarly, the performance with different numbers of updates T and the step size $\lambda$ are provided. Additionally, we also study the impact of number of neighbors N. We observe that on rOxford5K and rParis6K, different $\lambda$ has small impact on the performance. $T$=1 leads to the best performance, increasing $T$ is more computational expensive but the performances are not improved. $N$ is important

35th Conference on Neural Information Processing Systems (NeurIPS 2021).

in SSR, larger $N$ (*e.g.* $N = 400$) provides clear boost across different datasets. The final performances reported in the paper are bold numbers indicated in the table.

| $T$ | $\lambda$ | $N$ | rOxford5K [13, 7], mAP | | | rParis6K [13, 7], mAP | | |
|---|---|---|---|---|---|---|---|---|
| | | | Medium | Hard | Average | Medium | Hard | Average |
| Feature only [7] | | | 67.3 | 44.3 | 55.8 | 80.6 | 61.5 | 71.1 |
| 1 | 1e-4 | 100 | 72.1 | 49.4 | 60.8 | 81.2 | 63.0 | 72.1 |
| | 2e-4 | 100 | 72.3 | 49.4 | 60.9 | 81.3 | 63.2 | 72.3 |
| | 5e-4 | 100 | 72.8 | 50.6 | 61.7 | 81.0 | 62.8 | 71.9 |
| | 1e-4 | 200 | 70.7 | 52.1 | 61.4 | 81.6 | 64.3 | 73.0 |
| | 2e-4 | 200 | 71.7 | 50.0 | 60.9 | 81.6 | 64.2 | 72.9 |
| | 5e-4 | 200 | 73.3 | 52.8 | 63.1 | 81.5 | 64.1 | 72.8 |
| | 1e-4 | 300 | 72.2 | 51.7 | 62.0 | 82.8 | 65.8 | 74.3 |
| | 2e-4 | 300 | 73.5 | 54.3 | 63.9 | 79.1 | 60.9 | 70 .0 |
| | 5e-4 | 300 | 72.1 | 53.7 | 62.9 | 79.8 | 61.7 | 70.8 |
| | 1e-4 | 400 | 72.2 | 52.1 | 62.2 | 81.5 | 63.6 | 72.6 |
| | 2e-4 | 400 | 71.6 | 49.6 | 60.6 | **84.4** | **67.8** | **76.1** |
| | 5e-4 | 400 | 72.9 | 52.8 | 62.9 | 80.6 | 63.0 | 71.8 |
| | 1e-4 | 500 | 71.5 | 51.5 | 61.5 | 83.0 | 67.0 | 75.0 |
| | 2e-4 | 500 | 72.3 | 50.9 | 61.6 | 82.7 | 66.4 | 74.6 |
| | 5e-4 | 500 | 68.9 | 49.2 | 59.1 | 83.1 | 67.1 | 75.1 |
| 2 | 1e-4 | 100 | 71.3 | 50.0 | 60.7 | 80.4 | 62.1 | 71.3 |
| | 2e-4 | 100 | 72.4 | 50.9 | 61.7 | 81.2 | 63.1 | 72.2 |
| | 5e-4 | 100 | 72.5 | 50.3 | 61.4 | 81.2 | 63.0 | 72.1 |
| | 1e-4 | 200 | 72.7 | 52.8 | 62.8 | 79.1 | 61.1 | 70.1 |
| | 2e-4 | 200 | 73.9 | 52.0 | 63.0 | 81.1 | 63.9 | 72.5 |
| | 5e-4 | 200 | 74.0 | 54.0 | 64 | 79.5 | 62.0 | 70.8 |
| | 1e-4 | 300 | 73.3 | 51.6 | 62.5 | 78.8 | 59.3 | 69.1 |
| | 2e-4 | 300 | 73.5 | 54.3 | 63.9 | 79.1 | 60.9 | 70.0 |
| | 5e-4 | 300 | 73.4 | 54.2 | 63.8 | 77.5 | 58.7 | 68.1 |
| | 1e-4 | 400 | **75.6** | **54.2** | **64.9** | 82.3 | 64.2 | 73.3 |
| | 2e-4 | 400 | 73.1 | 50.7 | 61.9 | 78.6 | 59.5 | 69.1 |
| | 5e-4 | 400 | 71.5 | 50.6 | 61.1 | 79.5 | 60.3 | 69.9 |
| | 1e-4 | 500 | | | | | | |
| | 2e-4 | 500 | | | | | | |
| | 5e-4 | 500 | | | | | | |

**Table 1: Hyper-parameters analysis of SSR on rOxford5K and rParis6K.** We report results on rOxford5K [13] and rParis6K [13] using features in [7]. Note that **bold** numbers are reported in the paper.

| QE | No. neighbors in QE methods | rOxford5K [13, 7], mAP | | | rParis6K [13, 7], mAP | | |
|---|---|---|---|---|---|---|---|
| | | Medium | Hard | Average | Medium | Hard | Average |
| Feature only [7] | | 67.3 | 44.3 | 55.8 | 80.6 | 61.5 | 71.1 |
| AQE [2] | 1 | **70.8** | **48.0** | **59.4** | 82.1 | 64.4 | 73.3 |
| AQE [2] + SSR | 1 | **74.2** | **54.4** | **64.3** | 81.7 | 65.0 | 73.4 |
| AQE [2] | 3 | 68.2 | 44.0 | 56.1 | 80.5 | 61.8 | 71.2 |
| AQE [2] + SSR | 3 | 71.6 | 51.1 | 61.4 | 80.0 | 62.8 | 71.4 |
| AQE [2] | 5 | 65.8 | 41.7 | 53.8 | 84.1 | 67.2 | 75.7 |
| AQE [2] + SSR | 5 | 68.0 | 47.6 | 57.8 | 83.7 | 67.6 | 75.7 |
| AQE [2] | 7 | 65.0 | 41.0 | 53 | 84.6 | 67.8 | 76.2 |
| AQE [2] + SSR | 7 | 67.5 | 47.0 | 57.3 | 84.2 | 68.2 | 76.2 |
| AQE [2] | 9 | 64.3 | 41.2 | 52.8 | **85.3** | **68.8** | **77.1** |
| AQE [2] + SSR | 9 | 68.2 | 48.2 | 58.2 | **84.5** | **68.7** | **76.6** |
| AQEwD [6] | 1 | **70.8** | **48.0** | **59.4** | 81.9 | 63.9 | 72.9 |
| AQEwD [6] + SSR | 1 | **74.1** | **54.1** | **64.1** | 81.5 | 64.8 | 73.2 |
| AQEwD [6] | 3 | 71.9 | 48.7 | 60.3 | 82.2 | 65.3 | 73.8 |
| AQEwD [6] + SSR | 3 | 75.6 | 56.1 | 65.9 | 82.7 | 66.1 | 74.4 |
| AQEwD [6] | 5 | 72.2 | 48.8 | 60.5 | 83.5 | 66.3 | 74.9 |
| AQEwD [6] + SSR | 5 | 74.2 | 53.8 | 64.0 | 83.3 | 67.0 | 75.2 |
| AQEwD [6] | 7 | 71.4 | 46.5 | 59.0 | 84.0 | 67.0 | 75.5 |
| AQEwD [6] + SSR | 7 | 69.2 | 48.9 | 59.1 | 83.7 | 67.5 | 75.6 |
| AQEwD [6] | 9 | 66.0 | 42.7 | 54.4 | **84.5** | **67.6** | **76.1** |
| AQEwD [6] + SSR | 9 | 68.7 | 48.5 | 58.6 | **83.9** | **67.9** | **75.9** |
| $\alpha$QE [14] | 1 | 68.3 | 45.5 | 56.9 | 81.4 | 63.0 | 72.2 |
| $\alpha$QE [14] + SSR | 1 | 71.6 | 52.5 | 62.1 | 81.2 | 64.3 | 72.8 |
| $\alpha$QE [14] | 3 | 69.0 | 45.8 | 57.4 | 82.3 | 64.4 | 73.4 |
| $\alpha$QE [14] + SSR | 3 | 72.8 | 53.5 | 63.2 | 81.8 | 65.2 | 73.5 |
| $\alpha$QE [14] | 5 | **69.5** | **46.1** | **57.8** | 83.0 | 65.5 | 74.3 |
| $\alpha$QE [14] + SSR | 5 | **73.6** | **53.7** | **63.7** | 82.3 | 66.0 | 74.2 |
| $\alpha$QE [14] | 7 | 73.2 | 52.5 | 62.9 | 83.5 | 66.2 | 74.9 |
| $\alpha$QE [14] + SSR | 7 | 73.2 | 52.5 | 62.9 | 82.7 | 66.5 | 74.6 |
| $\alpha$QE [14] | 9 | 70.0 | 46.8 | 58.4 | **84.0** | **67.0** | **75.5** |
| $\alpha$QE [14] + SSR | 9 | 73.3 | 51.8 | 62.6 | **83.1** | **67.0** | **75.1** |
| DQE [1] | 1 | **68.2** | **44.0** | **56.1** | 80.5 | 61.8 | 71.2 |
| DQE [1] + SSR | 1 | **71.6** | **51.1** | **61.4** | 80.0 | 62.8 | 71.4 |
| DQE [1] | 3 | 66.4 | 39.9 | 53.2 | 81.7 | 63.4 | 72.6 |
| DQE [1] + SSR | 3 | 65.0 | 43.2 | 54.1 | 81.8 | 65.1 | 73.5 |
| DQE [1] | 5 | 66.6 | 42.3 | 54.4 | 83.1 | 65.4 | 74.3 |
| DQE [1] + SSR | 5 | 64.5 | 43.1 | 53.8 | 82.5 | 65.9 | 74.2 |
| DQE [1] | 7 | 63.1 | 38.6 | 50.9 | 83.6 | 66.2 | 74.9 |
| DQE [1] + SSR | 7 | 64.3 | 42.4 | 53.4 | 83.2 | 66.8 | 75.0 |
| DQE [1] | 9 | 63.9 | 40.2 | 52.1 | **84.3** | **67.2** | **75.8** |
| DQE [1] + SSR | 9 | 64.3 | 42.6 | 53.5 | **83.5** | **67.3** | **75.4** |

Table 2: **Hyper-parameter analysis of QE and QEs + SSR on rOxford5K and rParis6K.** We report mAP on rOxford5K [13] and rParis6K [13]. The results are with features in [7]. Note that the **bold** numbers are reported in the paper.

## 2.2 Analysis of hyper-parameters in QE methods on image retrieval benchmarks

To combine QE and SSR, we directly apply SSR to the retrieved samples given by QE. SSR is trained with using the hard training data sampling described in the previous section. The results are present in Table 2. As we can see, in most cases, our SSR can again improve the performance of QEs. The improvement is also robust with respect to different hyper-parameters of QEs.

| $k_1$ | $k_2$ | $\lambda$ | SSR | rOxford5K [13, 7], mAP | | | rParis6K [13, 7], mAP | | |
|---|---|---|---|---|---|---|---|---|---|
| | | | | Medium | Hard | Average | Medium | Hard | Average |
| Feature only [7] | | | | 67.3 | 44.3 | 55.8 | 80.6 | 61.5 | 71.1 |
| 40 | 20 | 0.1 | | 72.2 | 52.8 | 62.5 | 83.7 | 66.8 | 75.3 |
| | | 0.1 | ✓ | 71.7 | 52.7 | 62.2 | 84.6 | 68.2 | 76.4 |
| | | 0.3 | | 65.6 | 44.1 | 54.9 | 86.5 | 71.5 | 79 .0 |
| | | 0.3 | ✓ | 73.5 | 53.0 | 63.3 | 87.3 | 72.3 | 79.8 |
| | | 0.5 | | 61.7 | 35.2 | 48.5 | 87.8 | 74.7 | 81.3 |
| | | 0.5 | ✓ | 72.3 | 53.1 | 62.7 | 88.8 | 75.8 | 82.3 |
| 80 | 40 | 0.1 | | 73.8 | 55.9 | 64.9 | 83.7 | 67.0 | 75.4 |
| | | 0.1 | ✓ | 72.9 | 52.5 | 62.7 | 84.5 | 68.3 | 76.4 |
| | | 0.3 | | 70.3 | 47.1 | 58.7 | 86.8 | 72.2 | 79.5 |
| | | 0.3 | ✓ | 74.3 | 52.9 | 63.6 | 87.5 | 73.0 | 80.3 |
| | | 0.5 | | 65.4 | 41.4 | 53.4 | **87.9** | **74.8** | **81.4** |
| | | 0.5 | ✓ | 73.5 | 54.3 | 63.9 | **88.7** | **75.9** | **82.3** |
| 160 | 80 | 0.1 | | 73.8 | 55.3 | 64.6 | 83.2 | 66.3 | 74.8 |
| | | 0.1 | ✓ | 73.9 | 53.3 | 63.6 | 83.9 | 67.4 | 75.7 |
| | | 0.3 | | **72.1** | **50.7** | **61.4** | 86.7 | 72.3 | 79.5 |
| | | 0.3 | ✓ | **75.0** | **53.1** | **64.1** | 87.3 | 73.0 | 80.2 |
| | | 0.5 | | 68.3 | 47.0 | 57.7 | 87.8 | 74.6 | 81.2 |
| | | 0.5 | ✓ | 73.7 | 52.9 | 63.3 | 88.4 | 75.5 | 82.0 |

**Table 3: Hyper-parameters analysis of k-reciprocal [20] and k-reciprocal [20] + SSR on image retrieval benchmark.** We report mAP on rOxford5K [13] and rParis6K [13]. The results are with features in [7]. Note that the **bold** numbers are reported in the paper.

## 2.3   Analysis of hyper-parameters in k-reciprocal [20] on rOxford5K and rParis6K

To combine k-reciprocal [20] and SSR, we directly apply SSR to the retrieved samples given by k-reciprocal [20]. Similarly, SSR is trained with using the hard training data sampling described in the previous section. The results are present in Table 3. As we can see, our SSR can further improve the best performance (**bold** numbers) obtained by k-reciprocal [20]. The improvement is also robust with respect to different hyper-parameters of QEs.

## 3   Detailed results of CUB, CARS and SOP

In this section, we provide detailed results on image retrieval benchmark. The results are illustrated in Table 4. For each feature, we report recall at 1 (R@1), precision at R (PR) and mean average precision at R (mAP@R).

We also report results with different numbers of neighbors for approaches which consist of applying our method on features (*'Feature + SSR'*) and combining our method with k-reciprocal [20] (*'Feature + k-reciprocal [20] + SSR'*). From the results, we can see, first, the re-ranking approaches studied in the paper (query expansion, k-reciprocal [20] and our approach) can improve metric mAP@R and PR, and keep the same level of performance in terms of recall. Second, our approach is robust with respect to the number of neighbors (N). N=200 leads to the most stable performance on CUB [18] and CARS [9] and N=50 is optimal for SOP [?]. We thus conducted experiments with N = 200 on CUB [18] and CARS [9] and N = 50 on SOP [?] for combining query expansion and our approach.

| | CUB200 [18] | | | CARS196 [9] | | | SOP [?] | | |
|---|---|---|---|---|---|---|---|---|---|
| | R@1 ↑ | PR ↑ | MAP@R ↑ | R@1 ↑ | PR ↑ | MAP@R ↑ | R@1 ↑ | PR ↑ | MAP@R ↑ |
| **Feature Only** | | | | | | | | | |
| GL [3] | 65.4 | 35.6 | 24.5 | 85.3 | 37.8 | 27.8 | 75.7 | 49.9 | 46.9 |
| PA [?] | 68.5 | 37.6 | 27.0 | 86.0 | 37.8 | 28.3 | 78.5 | 54.0 | 51.0 |
| PNCA++ [16] | 71.7 | 40.5 | 29.6 | 89.8 | 42.3 | 33.2 | 81.5 | 58.3 | 55.4 |
| **Feature + SSR** | | | | | | | | | |
| GL [3] + SSR (N = 50) | 65.1 | 35.7 | 26.9 | 80.7 | 37.8 | 29.8 | 76.3 | 53.5 | 50.8 |
| GL [3] + SSR (N = 100) | 65.3 | 40.2 | 31.5 | 83.1 | 40.0 | 34.0 | 76.3 | 53.6 | 50.9 |
| GL [3] + SSR (N = 200) | 65.2 | 43.2 | 34.0 | 83.2 | 45.4 | 38.3 | 76.4 | 53.0 | 50.1 |
| GL [3] + SSR (N = 300) | 64.0 | 43.2 | 33.8 | 77.8 | 46.5 | 38.8 | - | - | - |
| PA [?] + SSR (N = 50) | 66.5 | 37.6 | 29.1 | 82.4 | 37.8 | 30.1 | 78.4 | 57.3 | 54.8 |
| PA [?] + SSR (N = 100) | 66.5 | 42.1 | 33.8 | 84.6 | 39.8 | 33.9 | 78.3 | 57.3 | 54.6 |
| PA [?] + SSR (N = 200) | 64.5 | 44.4 | 35.5 | 81.6 | 44.8 | 38.1 | 78.3 | 56.7 | 54.0 |
| PA [?] + SSR (N = 300) | 63.1 | 44.4 | 35.3 | 80.5 | 45.9 | 38.7 | - | - | - |
| PNCA++ [16] + SSR (N = 50) | 70.2 | 40.5 | 32.1 | 87.2 | 42.4 | 35.2 | 81.7 | 62.9 | 60.6 |
| PNCA++ [16] + SSR (N = 100) | 71.3 | 46.1 | 38.11 | 88.3 | 44.7 | 39.4 | 81.5 | 63.1 | 60.7 |
| PNCA++ [16] + SSR (N = 200) | 66.2 | 47.9 | 39.5 | 86.3 | 50.9 | 45.1 | 81.9 | 61.6 | 59.0 |
| PNCA++ [16] + SSR (N = 300) | 64.3 | 47.3 | 38.8 | 82.7 | 52.4 | 45.8 | - | - | - |
| **Feature + Query Expansion** | | | | | | | | | |
| GL [3] + AQE [2] | 64.4 | 38.3 | 28.3 | 80.9 | 43.3 | 34.5 | 75.7 | 51.4 | 49.2 |
| GL [3] + AQEwD [6] | 65.6 | 38.4 | 28.3 | 83.9 | 43.4 | 34.6 | 75.7 | 51.8 | 49.3 |
| GL [3] + DQE [1] | 61.3 | 36.7 | 26.3 | 81.0 | 41.1 | 31.6 | 75.7 | 50.9 | 48.7 |
| GL [3] + αQE [14] | 65.4 | 38.3 | 28.1 | 82.5 | 43.2 | 34.3 | 75.7 | 51.3 | 49.0 |
| PA [?] + AQE [2] | 66.8 | 40.6 | 31.2 | 83.2 | 43.1 | 34.9 | 79.2 | 56.9 | 54.8 |
| PA [?] + AQEwD [6] | 69.1 | 40.8 | 31.2 | 86.0 | 42.9 | 34.5 | 79.1 | 57.4 | 55.0 |
| PA [?] + DQE [1] | 64.6 | 40.4 | 30.7 | 81.6 | 43.0 | 34.4 | 79.1 | 57.1 | 55.0 |
| PA [?] + αQE [14] | 68.0 | 40.6 | 30.9 | 85.3 | 42.7 | 34.3 | 79.2 | 56.7 | 54.3 |
| PNCA++ [16] + AQE [2] | 69.9 | 43.6 | 34.1 | 85.0 | 48.7 | 40.9 | 81.5 | 60.5 | 58.4 |
| PNCA++ [16] + AQEwD [6] | 70.7 | 43.6 | 34.2 | 88.2 | 48.7 | 40.9 | 81.4 | 61.0 | 58.6 |
| PNCA++ [16] + DQE [1] | 70.0 | 42.9 | 33.0 | 86.9 | 45.0 | 37.0 | 81.4 | 60.4 | 58.3 |
| PNCA++ [16] + αQE [14] | 68.2 | 43.6 | 34.1 | 86.8 | 48.6 | 40.9 | 81.5 | 60.4 | 58.4 |
| **Feature + Query Expansion + SSR** | | | | | | | | | |
| GL [3] + AQE +SSR | 62.1 | 43.4 | 34.1 | 75.1 | 48.1 | 40.3 | 75.6 | 52.2 | 49.5 |
| GL [3] + AQEwD +SSR | 60.4 | 43.1 | 34.0 | 71.3 | 47.5 | 39.2 | 75.8 | 52.6 | 49.9 |
| GL [3] + DQE +SSR | 60.8 | 42.5 | 33.1 | 75.6 | 47.0 | 39.4 | 75.6 | 52.5 | 49.8 |
| GL [3] + αQE +SSR | 62.6 | 43.0 | 33.6 | 73.8 | 48.3 | 40.4 | 75.6 | 52.4 | 49.7 |
| PA [?] + AQE +SSR | 64.5 | 45.1 | 36.1 | 79.4 | 47.0 | 39.5 | 77.6 | 55.4 | 53.8 |
| PA [?] + AQEwD +SSR | 63.0 | 44.7 | 35.9 | 80.3 | 47.1 | 39.7 | 78.3 | 56.9 | 54.2 |
| PA [?] + DQE +SSR | 63.2 | 44.7 | 35.5 | 76.5 | 46.5 | 39.0 | 77.8 | 56.2 | 53.8 |
| PA [?] + αQE +SSR | 62.8 | 43.9 | 34.7 | 76.4 | 44.5 | 35.7 | 78.1 | 56.6 | 54.0 |
| PNCA++ [16] + AQE +SSR | 66.4 | 48.9 | 40.2 | 83.4 | 54.5 | 48.3 | 81.1 | 62.7 | 60.5 |
| PNCA++ [16] + AQEwD +SSR | 68.6 | 49.5 | 41.0 | 85.3 | 54.4 | 48.4 | 81.3 | 62.8 | 60.5 |
| PNCA++ [16] + DQE +SSR | 65.7 | 48.6 | 39.9 | 83.7 | 52.4 | 46.2 | 81.3 | 62.7 | 60.4 |
| PNCA++ [16] + αQE +SSR | 66.8 | 49.2 | 40.7 | 82.6 | 54.6 | 48.4 | 81.2 | 62.7 | 60.4 |
| **Feature + k-reciprocal [20]** | | | | | | | | | |
| GL [3] + [20] | 65.6 | 46.9 | 37.6 | 84.9 | 56.9 | 49.9 | 72.4 | 54.3 | 51.7 |
| PA [?] + [20] | 68.3 | 50.5 | 41.8 | 86.0 | 56.6 | 50.2 | 75.5 | 58.7 | 56.3 |
| PNCA++ [16]+ [20] | 73.6 | 55.9 | 48.1 | 90.4 | 63.8 | 58.5 | 78.9 | 64.0 | 61.7 |
| **Feature + k-reciprocal [20] + SSR** | | | | | | | | | |
| GL [3] + [20] + SSR (N = 50) | 66.4 | 46.9 | 37.8 | 84.1 | 56.9 | 50.0 | 73.8 | 55.1 | 52.5 |
| GL [3] + [20] + SSR (N = 100) | 66.5 | 46.9 | 38.2 | 84.9 | 56.9 | 49.9 | 73.6 | 55.0 | 52.3 |
| GL [3] + [20] + SSR (N = 200) | 66.2 | 47.4 | 38.7 | 84.2 | 57.3 | 51.1 | 73.0 | 54.7 | 51.9 |
| GL [3] + [20] + SSR (N = 300) | 66.3 | 47.5 | 38.8 | 84.9 | 56.9 | 49.9 | | | |
| PA [?]+ [20] + SSR (N = 50) | 68.3 | 50.5 | 41.8 | 85.7 | 56.6 | 50.2 | 75.6 | 59.1 | 56.6 |
| PA [?]+ [20] + SSR (N = 100) | 69.0 | 50.7 | 42.3 | 86.0 | 56.6 | 50.2 | 75.2 | 58.8 | 56.3 |
| PA [?]+ [20] + SSR (N = 200) | 68.7 | 50.7 | 42.3 | 86.1 | 57.0 | 50.9 | 75.3 | 58.6 | 56.2 |
| PA [?]+ [20] + SSR (N = 300) | 68.3 | 50.3 | 42.0 | 85.8 | 57.1 | 50.9 | | | |
| PNCA++ [16]+ [20] + SSR (N = 50) | 73.6 | 55.9 | 48.1 | 90.4 | 63.9 | 58.5 | 79.9 | 64.6 | 62.3 |
| PNCA++ [16]+ [20] + SSR (N = 100) | 74.1 | 56.5 | 48.8 | 90.4 | 63.8 | 58.5 | 79.6 | 64.4 | 62.1 |
| PNCA++ [16]+ [20] + SSR (N = 200) | 74.3 | 57.3 | 49.8 | 90.0 | 65.0 | 60.4 | 79.7 | 64.2 | 61.9 |
| PNCA++ [16]+ [20] + SSR (N = 300) | 74.1 | 56.3 | 48.9 | - | - | - | - | - | - |

Table 4: **Image retrieval:** Detailed results on CUB [18], CARS[9], SOP [?]. We report Recall at 1 (R@1), PR and mAP@R for all the methods and datasets.

# 4 Analysis of hyper-parameters in query expansion methods on CUB, CARS and SOP

In this section, we provide an analysis on hyper-parameters of query expansion methods. All the approaches are explained in. For AQE and AQEwD, the only hyper-parameter is the number of neighbor, while in alphaQE and DQE, there exists an additional hyper-parameter: alpha for alphaQE and the number of negative samples in DQE. The plots of different approaches as well as different alpha and numbers of negative samples are in Figure 1. We can see that the most crucial hyper-parameter remains the number of neighbors. Moreover, large number of neighbors degrades the performances for all the datasets and features.

We thus set $alpha = 1$ for alphaQE and number of negative samples as 200 in our experiments. The detailed results are provided in Table 5. The final number of neighbors we used are shown in bold in Table 5.

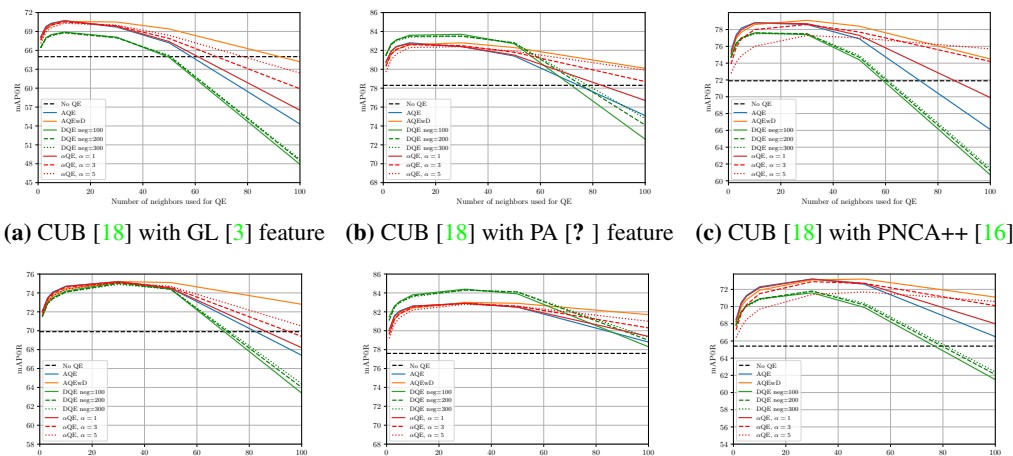

**(a)** CUB [18] with GL [3] feature  **(b)** CUB [18] with PA [?] feature  **(c)** CUB [18] with PNCA++ [16]

**(d)** CARS [9] with GL [3] feature  **(e)** CARS [9] with PA [?] feature  **(f)** CARS [9] with PNCA++ [16]

**Figure 1:** Ablation study on the hyper-parameters of four query expansion baselines on CUB [18] and CARS [9] with different features.

| neighbors | CUB | | | | CARS | | | | SOP | | | |
|---|---|---|---|---|---|---|---|---|---|---|---|---|
| | AQE | AQEwD | DQE | alphaQE | AQE | AQEwD | DQE | alphaQE | AQE | AQEwD | DQE | alphaQE |
| **GroupLoss** | | | | | | | | | | | | |
| 1 | 68.2 | 67.7 | 66.4 | 68.0 | 72.1 | 71.7 | 71.5 | 72.0 | **56.6** | 56.2 | **55.9** | **56.6** |
| 3 | 69.8 | 69.3 | 67.9 | 69.7 | 73.5 | 73.1 | 72.9 | 73.4 | 54.6 | **56.7** | 54.4 | 54.7 |
| 5 | 70.3 | 70.0 | 68.4 | 70.2 | 74.1 | 73.7 | 73.4 | 74.0 | 50.6 | 55.6 | 51.2 | 50.8 |
| 10 | **70.7** | **70.6** | **68.8** | **70.7** | 74.7 | 74.4 | 74.1 | 74.7 | – | – | – | – |
| 30 | 69.7 | 70.5 | 68.0 | 69.8 | **75.2** | **75.2** | **75.0** | **75.2** | – | – | – | – |
| 50 | 67.2 | 69.4 | 65.1 | 67.4 | 74.5 | 75.1 | 74.4 | 74.5 | – | – | – | – |
| 100 | 54.3 | 64.2 | 48.5 | 56.5 | 67.4 | 72.8 | 64.0 | 68.2 | – | – | – | – |
| **ProxyAnchor** | | | | | | | | | | | | |
| 1 | 80.8 | 80.4 | 81.5 | 80.7 | 80.1 | 79.7 | 81.1 | 80.0 | **80.4** | 80.2 | **81.0** | **80.4** |
| 3 | 81.9 | 81.6 | 82.6 | 81.9 | 81.6 | 81.2 | 82.5 | 81.5 | 78.8 | **80.9** | 80.7 | 79.0 |
| 5 | 82.4 | 82.1 | 83.0 | 82.4 | 82.1 | 81.8 | 83.0 | 82.0 | 74.9 | 80.1 | 78.9 | 75.2 |
| 10 | **82.8** | 82.6 | **83.5** | **82.7** | 82.6 | 82.4 | 83.7 | 82.6 | – | – | – | – |
| 30 | 82.4 | **82.8** | 83.5 | 82.4 | **82.9** | **83.0** | **84.3** | **82.9** | – | – | – | – |
| 50 | 81.4 | 82.3 | 82.8 | 81.5 | 82.5 | 82.9 | 84.0 | 82.5 | – | – | – | – |
| 100 | 75.1 | 80.1 | 74.1 | 76.7 | 78.8 | 81.9 | 79.0 | 79.4 | – | – | – | – |
| **ProxyNCA++** | | | | | | | | | | | | |
| 1 | 75.5 | 75.0 | 74.7 | 75.2 | 68.5 | 68.1 | 67.8 | 68.3 | **71.4** | 71.1 | **71.2** | **71.5** |
| 3 | 77.4 | 77.0 | 76.3 | 77.2 | 70.5 | 70.0 | 70.0 | 70.3 | 69.9 | **72.0** | 71.0 | 70.1 |
| 5 | 78.2 | 77.7 | 77.0 | 78.0 | 71.3 | 70.8 | 70.0 | 71.2 | 66.0 | 71.2 | 69.4 | 66.4 |
| 10 | **78.8** | 78.6 | **77.6** | **78.8** | 72.3 | 71.9 | 70.9 | 72.2 | – | – | – | – |
| 30 | 78.6 | **79.1** | 77.5 | 78.7 | **73.2** | 73.1 | **71.8** | **73.2** | – | – | – | – |
| 50 | 76.9 | 78.4 | 74.7 | 77.3 | 72.6 | **73.2** | 70.1 | 72.7 | – | – | – | – |
| 100 | 66.1 | 74.5 | 61.2 | 69.9 | 66.5 | 71.1 | 62.1 | 68.0 | – | – | – | – |

**Table 5:** Grid search on hyper-parameters of different query expansion baselines on **train set**. We report mAP@R for comparison. We set the number of negatives as 200 for DQE and $\alpha$ as 1 for alphaQE.

# 5 Analysis of hyper-parameters in k-reciprocal [20] on CUB and CARS

We also provide similar analysis for k-reciprocal [20]. We find that the optimal weight for the feature distance $\alpha$ is 0.1. The plots with different $k_1$, $k_2$ and $\alpha = 0.1$ are shown in Figure 2. We can see k-reciprocal [20] is robust to hyper-parameters and results with large range of $k_1$, $k_2$ leads to important performance boost comparing to the feature comparison baseline.

The detailed results are provided in Table 6 for GL [3] feature,Table 7 for PA [?] feature and Table 6 for PNCA++ [16] feature. The hyper-parameter we used are bold in each table. For SOP [?], due to computational limitation, we didn't compute grid search and conducted all experiments with using $k_1 = 10$, $k_2 = 4$ and $\alpha = 0.1$.

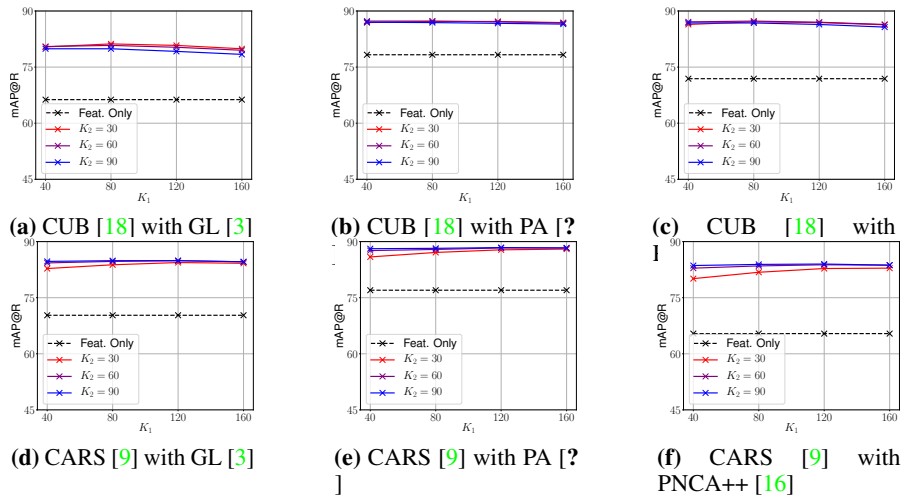

**(a)** CUB [18] with GL [3]    **(b)** CUB [18] with PA [?]    **(c)** CUB [18] with l

**(d)** CARS [9] with GL [3]    **(e)** CARS [9] with PA [?]    **(f)** CARS [9] with PNCA++ [16]

**Figure 2:** Analysis on hyper-parameters of k-reciprocal [20] on CUB [18] and CARS [9]. For each plot, we fix $\alpha = 0.1$

| k1 | k2 | CUB [18] | | | CARS [9] | | |
|---|---|---|---|---|---|---|---|
| | | $\alpha = 0.05$ | $\alpha = 0.10$ | $\alpha = 0.15$ | $\alpha = 0.05$ | $\alpha = 0.10$ | $\alpha = 0.15$ |
| 40 | 30 | 80.4 | 80.4 | 80.3 | 82.8 | 82.8 | 82.7 |
| | 60 | 80.5 | 80.5 | 80.4 | 84.3 | 84.3 | 84.2 |
| | 90 | 79.9 | 79.9 | 79.8 | 84.7 | 84.7 | 84.6 |
| | 120 | 78.8 | 78.8 | 78.7 | 84.8 | 84.8 | 84.7 |
| 80 | 30 | 81.2 | **81.2** | 81.1 | 83.9 | 83.8 | 83.8 |
| | 60 | 80.8 | 80.8 | 80.7 | 84.8 | 84.7 | 84.7 |
| | 90 | 79.9 | 79.9 | 79.8 | 84.9 | 84.9 | 84.9 |
| | 120 | 78.7 | 78.7 | 78.6 | 84.9 | 84.9 | 84.8 |
| 120 | 30 | 80.8 | 80.8 | 80.7 | 84.5 | 84.4 | 84.4 |
| | 60 | 80.3 | 80.3 | 80.2 | 84.9 | **84.9** | 84.9 |
| | 90 | 79.2 | 79.2 | 79.2 | 84.9 | 84.9 | 84.9 |
| | 120 | 78.0 | 78.0 | 78.0 | 84.8 | 84.8 | 84.7 |
| 160 | 30 | 79.9 | 79.9 | 79.9 | 84.2 | 84.2 | 84.2 |
| | 60 | 79.5 | 79.5 | 79.5 | 84.6 | 84.6 | 84.6 |
| | 90 | 78.4 | 78.4 | 78.4 | 84.6 | 84.6 | 84.5 |
| | 120 | 77.1 | 77.1 | 77.2 | 84.5 | 84.5 | 84.4 |

**Table 6:** K-reciprocal grid search on CUB [18] and CARS [9] with Group Loss feature [3], we report mAP@R on train sets.

| k1 | k2 | CUB [18] | | | CARS [9] | | |
|---|---|---|---|---|---|---|---|
| | | $\alpha = 0.05$ | $\alpha = 0.10$ | $\alpha = 0.15$ | $\alpha = 0.05$ | $\alpha = 0.10$ | $\alpha = 0.15$ |
| 40 | 30 | 86.9 | 86.9 | 86.9 | 85.9 | 85.9 | 85.9 |
| | 60 | 87.3 | 87.3 | 87.2 | 87.6 | 87.6 | 87.5 |
| | 90 | 87.0 | 87.0 | 87.0 | 88.2 | 88.1 | 88.1 |
| | 120 | 86.7 | 86.7 | 86.7 | 88.4 | **88.4** | 88.3 |
| 80 | 30 | 87.2 | 87.2 | 87.2 | 87.1 | 87.1 | 87.1 |
| | 60 | 87.3 | **87.3** | 87.3 | 88.0 | 87.9 | 87.9 |
| | 90 | 86.9 | 86.9 | 86.9 | 88.2 | 88.2 | 88.2 |
| | 120 | 86.6 | 86.6 | 86.6 | 88.4 | 88.3 | 88.3 |
| 120 | 30 | 87.2 | 87.2 | 87.2 | 87.8 | 87.8 | 87.8 |
| | 60 | 87.1 | 87.1 | 87.1 | 88.3 | 88.3 | 88.3 |
| | 90 | 86.7 | 86.7 | 86.8 | 88.4 | 88.4 | 88.3 |
| | 120 | 86.3 | 86.3 | 86.3 | 88.3 | 88.3 | 88.0 |
| 160 | 30 | 86.8 | 86.9 | 86.9 | 88.0 | 88.0 | 88.0 |
| | 60 | 86.8 | 86.8 | 86.8 | 88.3 | 88.3 | 88.3 |
| | 90 | 86.5 | 86.5 | 86.5 | 88.3 | 88.3 | 88.3 |
| | 120 | 86.0 | 86.0 | 86.0 | 88.1 | 88.1 | 88.1 |

**Table 7:** K-reciprocal grid search on CUB [18] and CARS [9] with Proxy Anchor feature [**?** ], we report mAP@R on train sets.

| k1 | k2 | CUB [18] | | | CARS [9] | | |
|---|---|---|---|---|---|---|---|
| | | $\alpha = 0.05$ | $\alpha = 0.10$ | $\alpha = 0.15$ | $\alpha = 0.05$ | $\alpha = 0.10$ | $\alpha = 0.15$ |
| 40 | 30 | 86.4 | 86.4 | 86.4 | 80.2 | 80.3 | 80.3 |
| | 60 | 87.1 | 87.1 | 87.1 | 82.9 | 82.9 | 82.8 |
| | 90 | 86.9 | 86.8 | 86.8 | 83.7 | 83.6 | 83.6 |
| | 120 | 86.6 | 86.5 | 86.5 | 84.0 | 83.9 | 83.8 |
| 80 | 30 | 87.2 | 87.1 | 87.1 | 81.8 | 81.8 | 81.9 |
| | 60 | 87.3 | **87.3** | 87.3 | 83.5 | 83.5 | 83.5 |
| | 90 | 86.8 | 86.8 | 86.8 | 83.9 | 83.9 | 83.8 |
| | 120 | 86.4 | 86.4 | 86.4 | 84.0 | 84.0 | 83.9 |
| 120 | 30 | 86.9 | 86.9 | 86.9 | 82.8 | 82.8 | 82.8 |
| | 60 | 86.9 | 87.0 | 86.9 | 83.8 | 83.8 | 83.8 |
| | 90 | 86.4 | 86.4 | 86.4 | 84.0 | **84.0** | 83.9 |
| | 120 | 85.9 | 85.9 | 85.9 | 83.9 | 83.9 | 83.9 |
| 160 | 30 | 86.3 | 86.3 | 86.3 | 82.8 | 82.9 | 82.9 |
| | 60 | 86.4 | 86.4 | 86.4 | 83.7 | 83.7 | 83.7 |
| | 90 | 85.7 | 85.7 | 85.8 | 83.8 | 83.7 | 83.7 |
| | 120 | 85.1 | 85.1 | 85.1 | 83.6 | 83.6 | 83.6 |

**Table 8:** K-reciprocal grid search on CUB [18] and CARS [9] with PNCA++ feature [16], we report mAP@R on train sets.

# 6 Dependency on the number of updates T for 1-shot classification on Mini-ImageNet [17]

We provide an analysis on the number of updates T. The results are in Figure 3 for 1-shot performance on Mini-ImageNet [17]. As claimed in the paper, we found the T=3 is the best for few-shot classification.

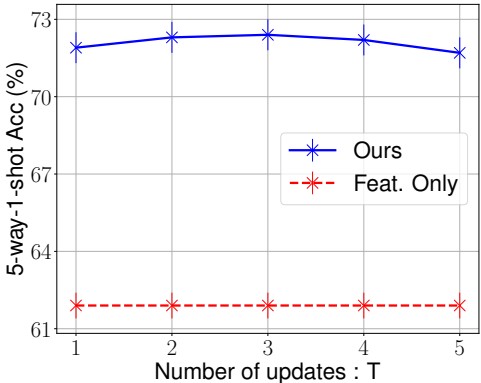

**Figure 3: 1-shot** performance on Mini-ImageNet [17] with WRN-28-10 [19]

# 7 Combining k-reciprocal [20] and our approach on transductive few-shot classification

In this section, we provide results of combining k-reciprocal [20] and our approach for transductive few-shot classification. We experimented on Mini-ImageNet [17], tiered-ImageNet [15] and CIFAR-FS [12] with the same architectures used in the paper: Conv-4-64, ResNet12, WRN-28-10. The results are shown in Table 9. First, both methods (our SSR and k-reciprocal [20] + SSR) consistently improve the feature comparison baseline. Second, our approach alone outperforms k-reciprocal [20] + SSR in most of the cases, which demonstrates the effect of our approach.

| Method | Mini-ImageNet [17] | tiered-ImageNet [15] | CIFAR-FS [12] |
|---|---|---|---|
| Conv-4-64 (Feat. Only) | 52.4±0.4 | 55.2±0.5 | 57.8±0.5 |
| Conv-4-64 + SSR | **62.1±0.6** | 65.1±0.6 | **72.0±0.6** |
| Conv-4-64 + k-reciprocal [20] + SSR | 60.8±0.6 | **65.9±0.6** | 69.5±0.6 |
| ResNet-12 (Feat. Only) | 57.6±0.5 | 68.8±0.5 | 66.4±0.5 |
| ResNet-12 + SSR | 68.1±0.6 | **81.2±0.6** | **76.8±0.6** |
| ResNet-12 + k-reciprocal [20] + SSR | **69.4±0.6** | 80.0±0.6 | 76.2±0.6 |
| WRN-28-10 (Feat. Only) | 61.9±0.5 | 69.4±0.5 | 69.5±0.5 |
| WRN-28-10 + SSR | **72.4±0.6** | **79.5±0.6** | **81.6±0.6** |
| WRN-28-10 + k-reciprocal [20] + SSR | 69.8±0.6 | 77.6±0.6 | 79.6±0.6 |

**Table 9:** Combining k-reciprocal [20] and our approach: 1-shot performance on mini-ImageNet [17], tiered-ImageNet [15] and CIFAR-FS [12]