# OpenReview forum: "Re-ranking for image retrieval and transductive few-shot classification"
_NeurIPS.cc/2021/Conference — NeurIPS 2021 Poster_

### Official Review · Reviewer_2iMx · 2021-07-01

**Rating:** 7
**Confidence:** 4

**Summary:**

The paper is about learning how to re-rank for retrieval with global image descriptors. There is a first step of retrieval to get the top-N neighbors to the query. Then, the NxN adjacency matrix is created which is the input for the re-ranking during training and during testing. The adjacency matrix is refined in an iterative way and by processing different sub-matrices each time.
During training the NxN adjacency matrix is forced to be close to the binary matrix that is contructed according to the class labels. At the same time, with the use of synthetic gradients, the backbone that is used to generate the feature vectors is updated so that the cosine similarity of the feature vectors facilitates the update of the adjacency matrix.
- this is a novel approach to re-ranking - the whole formulation is new for this task, so does the use of synthetic gradients
- the method is applied on two different tasks: image retrieval and transductive few-shot learning.

**Limitations And Societal Impact:**

This is not discussed at all

**Main Review:**

- it's refreshing to see such an approach which includes out of the box thinking to approach the task of re-ranking
- the ablations included in the paper are nicely conducted, they are informative and convincing
- the demonstrated improvements seem to be convincing.
- A missing bit is to see how does this compare with graph-based re-ranking such as  [10][22][75]. It is true that these methods carry a higher cost of having to maintain the graph of the whole dataset, while in the proposed case the NxN adjacency can be also computed on the fly. But anyways also the reciprocal neighbors need to carry out such a cost. This is one of the the main drawbacks of this work.
- what if the features backbone is not updated with synthetic gradients, but only gets updated according to the gradient of the loss in (6) and (7)? this is an interesting ablation which is missing
- (5) will accumulate many sub-matrices. The common row (eg in retrieval) is the row of the query. Which means that there are many summands (N actually) to get the result of this row. All other rows will appear only once. This feels that the refined sub-matrices should have much smaller values for the adjacency matrix to make sense. Do the authors verify something like that? This could be a main reason why more iterations do not work. Stopping after 1 iteration is probably still meaningful because the query row is used for re-ranking.
- "N = 50 for SOP, because they give the most stable  performance in “Feat. + Ours”": how these different values are chosen for different datasets? Is a proper validation set used? It is mentioned that the evaluation metric from [39] is used but not if the whole protocol is followed. [39] showed that prior work was tuning hyper-parameters on the test-sets which was leading to invalid comparisons.
- the training process results in a learned re-ranking process and update backbone. What is the effect of using only the updated backbone without any re-ranking? Is the updated backbone good to use for the similaritites of the adjacency matrix or also for the first step of retrieval to get the top-N results? Is it possible that the initial network (eg one from the group loss) is still beter for the initial ranking?
- how are the training batches of N samples generated? This information is missing.



------
post-rebuttal:
The authors provided a good response and I am still positive about the work
- there is significant difference between the proposed approach and GSS. GSS is learned and test on the same fixed dataset. This should be pointed out in the paper.
- [22] performs better but such an approach carries the cost of computing the graph for the whole dataset. this should be clarified and discussed.
- I now see the misunderstanding about updating the backbone - I agree that this should be properly clarified.
- The new results for smaller step size are helpful - I suggest to add in appendix.

**Time Spent Reviewing:**

4

---

> ### Author Response · Authors · 2021-08-10
> **Thanks for detailed review!**
>
> ### Thanks for detailed review! We address the concerns below.
>
> ### Q1: A missing bit is to see how does this compare with graph-based re-ranking such as [10][22][75]...
> We agree this could be an interesting comparison. We will add graph-based re-ranking baselines in the revised version.
>
> As a sanity check, according to [A], [22] can achieve `69.0, 89.5, 44.7, 80.0` for _rOxford5k medium_, _rParis6k medium_, _rOxford5k hard_, and _rParis6k hard_ respectively. Our SSR + k-reciprocal can achieve comparable performance `72.5, 88.2, 51.8, 75.0` with k1 = 160, k2 = 80, lambda=0.5 (please see the K-reciprocal table in [[Additional experiments]](https://openreview.net/forum?id=sneJD9juaNl&noteId=9asxKII1fr)).
>
> [A]: Radenovic et al, Revisiting Oxford and Paris: Large-Scale Image Retrieval Benchmarking, CVPR 2018.
>
> ### Q2: What if the features backbone is not updated with synthetic gradients...
> We should have made this more clear that only the features are updated not the backbone. We tried to update the backbone too, which doesn’t lead to significant improvement on the initial features and largely increases the memory overhead. We thus decided to present the current pipeline without end-to-end training of the backbone.
>
> Without updating features, the performance is worse. The ablation study can be found in Tab.3.a ('w.o. feat. update') and Tab.3.b ('w.o. feat. update'). We can see that updating features improves the performance by 1.4% on few-shot classification and by 3.8% for image retrieval.
>
> ### Q3: (5) will accumulate many sub-matrices. The common row (eg in retrieval) is the row of the query. Which means that there are many summands (N actually) to get the result of this row...
>
> Thanks for the suggestions. Since the similarities between one query and its nearest neighbors are always in the same position in the similarities matrix (2nd row), we hypothesize that the network can learn this scaling automatically. Empirically, we find that the predicted updates for query similarities (around 0.01) are much smaller than those of the database similarities (around 0.1)
>
>
> ### Q4: N = 50 for SOP, because they give the most stable  performance in “Feat. + Ours”": how these different values are chosen for different datasets?...
> The ablations on N are provided in table.2 in the supplementary material. As we would like to demonstrate performance over state-of-the-art features [3,5,11], we use the pre-trained features in [3,5,11] and follow the dataset split proposed by [3,5,11]. The main metric mAP@R is defined by [39] and it is the most relevant metric to image retrieval. We also report other metrics such as R@1 and PR in the supplementary material (table.2).
>
> ### Q5: the training process results in a learned re-ranking process and update backbone. What is the effect of using only the updated backbone without any re-ranking?…
> As explained in **Q2**, the backbone has not been updated. We will clarify this in the paper.
>
> ### Q6 : how are the training batches of N samples generated? This information is missing…
> We use the pre-trained feature backbone to compute the initial features for all retrieval candidates as well as the query. Then, the most similar $N$ samples to the query are selected in terms of the cosine similarity measure. We will clarify this in the revised version.
>
> ### We would be more than happy to discuss any further questions!

---

> > ### Comment · Reviewer_2iMx · 2021-08-26
> > **authors response is appreciated**
> >
> > thank you for the constructive response. I see the misunderstanding about updating the backbone - I agree that this should be properly clarified. The new results for smaller step size are helpful too - I suggest to include them in the appendix.
> > Regarding comparisons with [22]: it seems that [22] still performs the best. Nevertheless, it carries the extra cost of the graph over the whole dataset, I think this should be discussed in the paper.

---

> > > ### Author Response · Authors · 2021-08-27
> > > **Thank you for the suggestions**
> > >
> > > Thank you for the suggestions!
> > >
> > > We will add the experimental results of step size as well as the discussion to [22] to the final version of the paper.

---

> ### Author Response · Authors · 2021-08-26
> **Large step size is probably the reason why SSR stops after 1 iteration for image retrieval**
>
> The reason why the optimal step is 1 for image retrieval is probably due to the large step size $\lambda$ (c.f. equation 4, line 81). In the paper, we reported results with $\lambda=1e-3$ which is the same as SIB (c.f. line 171). We found that with using smaller $\lambda$ (5e-4, 2e-4), the optimal step is actually larger than 1. Here are the detailed results with $\lambda$ and nb of steps on rOxford and rParis.  We will update these results in the paper.
>
> | $\lambda$ | nStep | rOxford |       | rParis |       | avg   |
> |-----------|-------|---------|-------|--------|-------|-------|
> |           |       | Medium  | Hard  | Medium | Hard  |       |
> |           |       |         |       |        |       |       |
> | 2e-4      | 1     | 72.44   | 50.01 | 81.18  | 63.13 | 66.69 |
> | 2e-4      | 2     | 72.23   | 50.44 | 81.13  | 63.04 | 66.71 |
> | **2e-4**      | **3**     | 72.84   | 50.89 | 81.16  | 63.07 | **66.99** |
> |           |       |         |       |        |       |       |
> | 5e-4      | 1     | 72.40   | 50.44 | 80.99  | 62.86 | 66.67 |
> | **5e-4**      | **2**     | 73.01   | 50.90 | 81.21  | 63.11 | **67.06** |
> | 5e-4      | 3     | 72.43   | 50.97 | 80.88  | 62.67 | 66.74 |
> |           |       |         |       |        |       |       |
> | 1e-3      | 1     | 72.68   | 50.36 | 81.28  | 63.10 | 66.86 |
> | **1e-3**      | **2**     | 72.57   | 50.67 | 81.29  | 63.19 | **66.93** |
> | 1e-3      | 3     | 72.25   | 48.68 | 81.15  | 63.11 | 66.55 |
> |           |       |         |       |        |       |       |
> | **2e-3**      | **1**     | 71.58   | 49.01 | 81.29  | 63.11 | **66.77** |
> | 2e-3      | 2     | 72.23   | 50.44 | 81.13  | 63.04 | 66.71 |
> | 2e-3      | 3     | 72.19   | 49.2  | 80.78  | 62.45 | 66.16 |

---

### Official Review · Reviewer_wrsa · 2021-07-16

**Rating:** 7
**Confidence:** 1

**Summary:**

The paper deals with image retrieval, i.e. the problem of finding images in a database which are the most similar to a given query image. More specifically, the paper describes a method for re-ranking retrieval (and few-shot classification) results in order to improve performance. The technique consists in learning how to refine the retrieval results by first representing them as a similarity matrix, then learning a deep network G that learns how to produce a step that refines this matrix.

The contributions of the paper could be summarized as follows:

1) A novel method for re-ranking that uses a neural network to enhance the similarity matrix directly (which the authors call a ubgraph Similarity Refiner);
2) An approach for combining the method above with k-reciprocal re-ranking;
3) Experiments showing the method above leads to consistent improvements (although with varying levels of significance)

**Limitations And Societal Impact:**

Yes, the authors adequately addressed the limitations and potential negative societal impact of their work.

**Main Review:**

The paper reads well and has no typos. The motivation behind the paper is clear and the approach is interesting. The authors experiment on 6 different datasets comprised of both image-retrieval (CUBS, SOP, CARS) and few-shot classification problems (mini-ImageNet, tiered-ImageNet, CIFAR-FS). The authors provide some ablation studies decomposing some of their results to indicate how different steps of their method affect overall performance.

A significant part of the image retrieval literature have benchmarked other re-ranking strategies in the more challenging ROxford/RParis+1M datasets [A] instead of CUB/CARS/SOP. It would be interesting to know how this method would work in such a large-scale scenario.

[A]: Radenovic´ et al, Revisiting Oxford and Paris: Large-Scale Image Retrieval Benchmarking, CVPR 2018

# Very, very minor comments

- Typos: retrievel [l.59],
- Table 3 and 4 are wrapped around their respective paragraphs, probably due to size constraints, but still they seem a bit off. Their legend/subtitles are a bit packed making it a little bit confusing to read. But again, this is just a very minor comment.

**Time Spent Reviewing:**

8h

---

> ### Author Response · Authors · 2021-08-10
> **Thanks for the comments on our paper !**
>
> ### Thanks for the detailed comments! We will incorporate the suggested modifications in the revised version.
>
> ### Issue: A significant part of the image retrieval literature have benchmarked other re-ranking strategies in the more challenging ROxford/RParis+1M datasets [A] instead of CUB/CARS/SOP. It would be interesting to know how this method would work in such a large-scale scenario.
>
> As suggested, we add additional experiments on _rOxford5K_ and _rParis6K_ with all methods presented in the paper (QE, k-reciprocal and the proposed approach). However, due to the limited time for rebuttal, we were not able to include the 1M distractor set in our experiments. We will add a paragraph on the robustness of our method under distractors.
>
> Please refer to [[Additional experiments]](https://openreview.net/forum?id=sneJD9juaNl&noteId=9asxKII1fr) for more details.
>
> ### We would be more than happy to discuss any further questions!

---

> > ### Comment · Reviewer_wrsa · 2021-09-10
> > **Reviewer response**
> >
> > I've read the fellow reviewers reviews, their feedback, and the answers to my own concerns. I still think this is a good paper in the sense it proposes and tests a dynamic gnn that generalizes across datasets. The additional results reported by the authors on ROxford and RParis addressed some of my concerns. I am therefore keeping my initial rating of "7: good paper, accept".

---

> > > ### Author Response · Authors · 2021-09-11
> > > **Thank you!**
> > >
> > > Thank you so much for your comments and suggestions! They are the incentives to make our paper stronger.

---

### Official Review · Reviewer_STkv · 2021-07-16

**Rating:** 5
**Confidence:** 4

**Summary:**

This paper applies a type of graph neural network on top of fixed image representations to learn to refine similarity graphs for image retrieval and transductive few-shot classification. A contrastive loss over image pairs is used to optimize the model.


**Limitations And Societal Impact:**

Yes

**Main Review:**

Strengths
[S1] Experiments are shown for two different tasks: image retrieval and transductive few-shot classification with significant gains.
[S2] The proposed approach is shown to work in conjunction with parallel approaches for further performance improvements.

Weaknesses
[W1] The proposed approach to incorporate similarity matrix is a type of graph neural network applied on top of generated image features. This has been explored before for example in this CVPR paper [1] for image retrieval which this paper has not studied. This paper also fails to review any literature on graph neural networks.
[W2] Lack of clarity at several places. For example, it is not clear what is the operation "GraphSum(...)" [L108]. The final paragraph [L160-165] is also unclear: is the method using D^t for all t except for the last step t=T?
[W3] Most consistent performance improvements are shown for image retrieval. However, the image retrieval setup is not convincing. All of the methods in table 1 are adopted from different problems: AQE, DQE, and alphaQE test on the landmark retrieval datasets, while reciprocal-k is applied to the person re-id problem. It would make more sense to follow an existing retrieval pipeline for fair comparisons.

[1]Liu, Chundi, et al. "Guided similarity separation for image retrieval." Advances in Neural Information Processing Systems 32 (2019): 1556-1566.

**Time Spent Reviewing:**

5

---

> ### Author Response · Authors · 2021-08-10
> **Thanks for the detailed review!**
>
> ### Thanks for the detailed review! We address the concerns as follows.
>
> ## W1
> We agree that GNN-based and graph-based methods are related, will add a paragraph to summarize this line of research. We actually cite the work mentioned as [1] _Liu, Chundi et al., Guided Similarity Separation (GSS)_, as [33] in our paper as an example that exploits high-order neighbors. We will add the following discussion for the main differences between GSS and our SSR:
>
> Although both GSS [1] and SSR leverage the similarities between images to update features, the philosophies are different in the sense that
> 1. GSS, as well as many GNN based approaches, takes as input the adjacency matrix of the graph and updates only node/edge features. Note that the graph structure remains unchanged during the feedforward pass. In contrast, SSR doesn’t assume any predefined graph structure, and updates both the similarity matrix (i.e., the graph structure) and the image features in an alternating way (c.f equation 2 and 3). This makes SSR quite different from classical GNN-based approaches.
> 2. For a given database of images, e.g., _rOxford5K_, GSS would train a GNN based on the KNN graph of the entire database. The database dependent GNN is then used to perform image retrieval. This has the advantage of making full use of higher order neighborhood information of the database, but it becomes problematic if we want to expand the given database. In contrast, SSR is a meta-learning approach: it learns to re-rank the nearest neighbors of any query images (not the entire database). More specifically, every query image and its nearest neighbors define a task/episode and the re-ranking of the neighbors is done by using a meta-model (in our case, the network $G$).
>
> ## W2
> Please find below our clarifications.
> * $GraphSum$ is the aggregation of the predictions on all the subgraphs (c.f. line 106). Precisely, for a subgraph $M_i$, the update $g(M_i)$ only changes a portion of the similarity matrix $S$; different subgraph updates will sum up to obtain the update of $S$. Due to the page limit, we didn’t give a rigorous definition of $GraphSum$, which requires notations from graph theory. We will add a paragraph in the appendix introducing the full notations and giving an explicit formula.
> * [L160-165]: The Jaccard distance matrix $J$ is computed at the beginning, and it remains unchanged. At $t$-th iteration, $J$ is used to augment the updated similarity matrix $S^t$. The augmented similarity matrix is denoted as $D^t$ in L163. Our SSR takes as input $D^t$ and outputs the updates for $S^t$. By doing so, the only change is in equation 2, which is now $\tilde{S}^t = S^{t-1} + G(D^{t-1})$. Suppose we update $S$ for $T$ iterations, the final loss (i.e., equation 6) is computed using $S^T$. We’ll elaborate this in the revised version.
>
>
> ## W3
> We understand the concern on the image retrieval setup. The main reason that we didn't follow the classical setup is because SSR needs a train-set, which is not provided in the classical _rOxford5K_ and _rParis6K_ benchmarks. Instead, we turned to the metric learning setup by [39] and adopted the same datasets (i.e., _CUB_, _CAR_ and _SOP_ datasets) and evaluation metrics there. We argue that this choice is valid as image retrieval and person re-identification are typical applications of metric learning. This is also the reason that we brought different baselines (as pointed out, AQE, DQE, and alphaQE from landmark retrieval, while k-reciprocal is applied to person re-id) from different problems.
>
> We however agree that our argument would be stronger if we had experiments on other retrieval benchmarks. To this end, we add additional experiments on _rOxford5K_ and _rParis6K_ with all methods presented in the paper (QE, k-reciprocal and the proposed approach). Following [48], we take the _SfM120K_ dataset as the train-set. A similar conclusion can be drawn from these additional results, where a consistent improvement can be observed by updating the initial features using SSR. Please refer to [[Additional experiments]](https://openreview.net/forum?id=sneJD9juaNl&noteId=9asxKII1fr) for more details.
>
> [39] : Kevin Musgrave, Serge Belongie, and Ser-Nam Lim. A metric learning reality check. In ECCV, 2020.
>
> ### We would be more than happy to discuss any further questions!

---

### Author Response · Authors · 2021-08-10
**Additional experiments on rOxford5K and rParis6K**

### We thank all the reviewers for their valuable comments! We answer each reviewer's comments and provide here additional experiments requested by _Reviewer STkv_ and _Reviewer wrsa_.

## Additional experiments

We present new results on _rOxford5K_ and _rParis6K_. The main difficulty is that there is no standard clean training set for these two datasets. One choice is SFM120k used in [48], which is constructed with structure-from-motion pipeline, and clusters for the same 3D scene are cast as categories. We take the pre-trained resnet101 features used in [16] which are available at https://github.com/filipradenovic/cnnimageretrieval-pytorch.

We find that these features lead to excellent performance on the SFM120K dataset. For most training samples, the mAPs on the training set are already quite high. Thus, directly using the raw nearest neighbors for training our model makes it perform well only for high mAP queries. To address this problem, we sample only difficult examples: for each query, we sample 1K database images, the query and its nearest neighbors will be training samples only if the mAP is not saturated (<0.8) and there are sufficient true positive samples present in the nearest neighbors (>5).


Training SSR on these examples yields consistent improvements on _rOxford5K_ and _rParis6K_ for both medium and hard queries. The results are shown in the table below.

|                                       | rOxford5K |      | rParis6K |      |
|---------------------------------------|:---------:|:----:|:--------:|:----:|
|                                       |   Medium  | Hard |  Medium  | Hard |
|             Pretrained ResNet101 [16]            |    67.3   | 44.3 |   80.6   | 61.5 |
|                  [16]                 |   **73.4**    | 49.6 |   86.3   | 70.6 |
|                [22, A]               |    69.0   | 44.7 |   **89.5**   | **80.0** |
|         Pretrained ResNet101 + SSR (N = 100)         |    71.6   | 49.8 |   81.0   | 62.8 |
| Pretrained ResNet101 + K-reciprocal + SSR (N = 100) |    72.5   | **51.8** |   88.2   | 75.0 |


[A]: Radenovic et al, Revisiting Oxford and Paris: Large-Scale Image Retrieval Benchmarking, CVPR 2018.

As requested by  _Reviewer STkv_, **we also further investigate whether our model can improve the performance over query expansion (QE) and k-reciprocal methods.** In the following tables, we use the _same trained model_ as in table 1, and perform re-ranking on the top-100 retrieved images with different methods: ADE, ADEwD, AlphaQE, DQE and k-reciprocal.

Since there is no training and validation dataset for rOxford5K and rParis6K, we demonstrate our approach to be complementary to these methods, by reporting results obtained with different hyper-parameters for the different methods.

From the results, we can see our approach can improve the performance for most cases. Our model is also the one that produces the best results with well-chosen parameters :
- 75.0 improved from 71.9 on _rOxford5K medium_ with AQEwD (nb of neighbors = 3)
- 52.9 improved from 44.1 on _rOxford5K hard_ with K-reciprocal (k1=80, k2=40, lambda=0.1)
- 88.9 improved from 87.8 on _rParis6K medium_ with K-reciprocal (k1=160, k2=80, lambda=0.1)
- 75.8 improved from 74.7 on _rParis6K hard_ with K-reciprocal (k1=160, k2=80, lambda=0.1)

**AQE**

|   # neighbors  | rOxford5K |      | rParis6K |      |
|:--------------:|:---------:|:----:|:--------:|:----:|
|                |   Medium  | Hard |  Medium  | Hard |
| Raw feat. [16] |    67.3   | 44.3 |   80.6   | 61.5 |
|        1       |    70.8   | 48.0 |   82.1   | 64.4 |
|      +Ours     |    73.6   | 51.6 |   82.2   | 64.7 |
|        3       |    71.3   | 48.5 |   83.1   | 65.8 |
|      +Ours     |    73.0   | 50.8 |   83.1   | 65.9 |
|        5       |    65.8   | 41.7 |   84.1   | 67.2 |
|      +Ours     |    66.9   | 43.1 |   84.1   | 67.3 |
|        7       |    65.0   | 41.0 |   84.6   | 67.8 |
|      +Ours     |    67.3   | 43.6 |   84.4   | 67.8 |
|        9       |    64.3   | 41.2 |   85.3   | 68.8 |
|      +Ours     |    67.5   | 45.3 |   85.0   | 68.5 |
---

**AQEwD**

|   # neighbors  | rOxford5K |      | rParis6K |      |
|:--------------:|:---------:|:----:|:--------:|:----:|
|                |   Medium  | Hard |  Medium  | Hard |
| Raw feat. [16] |    67.3   | 44.3 |   80.6   | 61.5 |
|        1       |    70.8   | 48.0 |   81.9   | 63.9 |
|      +Ours     |    73.5   | 51.6 |   82.1   | 64.7 |
|        3       |    71.9   | 48.7 |   82.8   | 65.1 |
|      +Ours     |    75.0   | 52.5 |   82.9   | 65.5 |
|        5       |    72.2   | 48.8 |   83.5   | 66.3 |
|      +Ours     |    72.5   | 48.5 |   83.5   | 66.4 |
|        7       |    71.4   | 46.5 |   84.0   | 67.0 |
|      +Ours     |    69.2   | 46.0 |   83.9   | 67.0 |
|        9       |    66.0   | 42.7 |   84.5   | 67.6 |
|      +Ours     |    67.9   | 45.4 |   84.3   | 67.6 |
---

**AlphaQE**

|   # neighbors  | rOxford5K |      | rParis6K |      |
|:--------------:|:---------:|:----:|:--------:|:----:|
|                |   Medium  | Hard |  Medium  | Hard |
| Raw feat. [16] |    67.3   | 44.3 |   80.6   | 61.5 |
|        1       |    68.3   | 45.5 |   81.4   | 63.0 |
|      +Ours     |    72.5   | 50.2 |   81.8   | 64.1 |
|        3       |    69.0   | 45.8 |   82.3   | 64.4 |
|      +Ours     |    72.7   | 50.1 |   82.5   | 65.2 |
|        5       |    69.5   | 46.1 |   83.0   | 65.0 |
|      +Ours     |    73.2   | 50.5 |   83.1   | 66.0 |
|        7       |    69.7   | 46.5 |   83.5   | 66.2 |
|      +Ours     |    73.6   | 51.0 |   83.5   | 66.5 |
|        9       |    69.9   | 46.8 |   84.0   | 67.0 |
|      +Ours     |    72.9   | 49.7 |   83.9   | 67.1 |
---

**DQE**

|   # neighbors  | rOxford5K |      | rParis6K |      |
|:--------------:|:---------:|:----:|:--------:|:----:|
|                |   Medium  | Hard |  Medium  | Hard |
| Raw feat. [16] |    67.3   | 44.3 |   80.6   | 61.5 |
|        1       |    68.2   | 44.0 |   80.5   | 61.8 |
|      +Ours     |    71.4   | 48.6 |   80.8   | 62.7 |
|        3       |    65.4   | 39.7 |   81.7   | 63.4 |
|      +Ours     |    63.2   | 40.0 |   81.9   | 64.2 |
|        5       |    63.0   | 38.1 |   83.1   | 65.4 |
|      +Ours     |    63.4   | 40.0 |   83.1   | 65.6 |
|        7       |    63.9   | 40.1 |   83.6   | 66.2 |
|      +Ours     |    63.1   | 39.3 |   83.4   | 66.2 |
|        9       |    63.1   | 40.7 |   84.3   | 67.2 |
|      +Ours     |    65.0   | 41.6 |   84.2   | 67.0 |
---

**K-reciprocal**

|       K1       | K2 | lambda | rOxford5K |      | rParis6K |      |
|:--------------:|:--:|:------:|:---------:|:----:|:--------:|:----:|
|                |    |        |   Medium  | Hard |  Medium  | Hard |
| Raw feat. [16] |    |        |    67.3   | 44.3 |   80.6   | 61.5 |
|       40       | 20 |   0.1  |    72.2   | 52.8 |   83.7   | 66.8 |
|  +Ours         |    |        |    71.3   | 53.2 |   84.6   | 68.1 |
|       40       | 20 |   0.3  |    73.8   | 55.9 |   83.7   | 67.0 |
|  +Ours         |    |        |    71.3   | 51.1 |   84.3   | 67.6 |
|       40       | 20 |   0.5  |    73.8   | 55.3 |   83.2   | 66.3 |
|  +Ours         |    |        |    70.9   | 50.4 |   83.7   | 66.8 |
|       80       | 40 |   0.1  |    65.6   | 44.1 |   86.5   | 71.5 |
|  +Ours         |    |        |    74.4   | 52.9 |   87.1   | 71.9 |
|       80       | 40 |   0.3  |    70.3   | 47.1 |   86.8   | 72.2 |
|  +Ours         |    |        |    73.9   | 52.4 |   87.2   | 72.5 |
|       80       | 40 |   0.5  |    72.1   | 50.7 |   86.7   | 72.3 |
|      +Ours     |    |        |    73.0   | 51.2 |   87.0   | 72.3 |
|       160      | 80 |   0.1  |    61.7   | 35.2 |   87.8   | 74.7 |
|  +Ours         |    |        |    71.1   | 51.5 |   88.9   | 75.8 |
|       160      | 80 |   0.3  |    65.4   | 41.4 |   86.8   | 72.2 |
|  +Ours         |    |        |    72.3   | 52.3 |   87.2   | 72.5 |
|       160      | 80 |   0.5  |    68.3   | 47.0 |   87.8   | 74.6 |
|      +Ours     |    |        |    72.5   | 51.8 |   88.2   | 75.0 |
---

---

### Decision · Program_Chairs · 2021-09-27

**Decision:**

Accept (Poster)

**Comment:**

The paper presents an interesting approach for re-ranking based on learning similarity graph between data points to be ranked. Reviewers agree that the paper presents interesting ideas. A concern remains which is: comparison with our GNN based ranking methods. It would be important for authors to present strong GNN based ranking baselines.